# TeaMs-RL: Teaching LLMs to Generate Better Instruction Datasets via Reinforcement Learning

**Shangding Gu**                                                     *shangding.gu@berkeley.edu*
*UC Berkeley & Technical University of Munich*

**Alois Knoll**                                                              *knoll@in.tum.de*
*Technical University of Munich*

**Ming Jin**                                                                  *jinming@vt.edu*
*Virginia Tech*

**Reviewed on OpenReview:** *https://openreview.net/forum?id=YmwzfdJPXE*

## Abstract

The development of Large Language Models (LLMs) often confronts challenges stemming from the heavy reliance on human annotators in the reinforcement learning with human feedback (RLHF) framework, or the frequent and costly external queries tied to the self-instruct paradigm. In this work, we pivot to Reinforcement Learning (RL)—but with a twist. Diverging from the typical RLHF, which refines LLMs following instruction data training, we use RL to directly generate the foundational instruction dataset that alone suffices for fine-tuning. Our method, *TeaMs-RL*, uses a suite of textual operations and rules, prioritizing the diversification of training datasets. It facilitates the generation of high-quality data without excessive reliance on external advanced models, paving the way for a single fine-tuning step and negating the need for subsequent RLHF stages. Our findings highlight key advantages of our approach: reduced need for human involvement and fewer model queries (**only** $5.73\%$ **of the strong baseline's total**), along with enhanced capabilities of LLMs in crafting and comprehending complex instructions compared to strong baselines, and substantially improved model privacy protection. Code is available at the link: `https://github.com/SafeRL-Lab/TeaMs-RL`

## 1  Introduction

In the dynamic realm of Large Language Models (LLMs), there has been a pronounced migration of their capabilities into diverse sectors, from chat robots (OpenAI, 2023; Zhao et al., 2023a; Touvron et al., 2023a) and robotics (Ahn et al., 2022; Ren et al., 2023), to autonomous driving (Fu et al., 2023; Tang et al., 2023). Amidst this broad applicability, the capacity to train with targeted instructions and pertinent responses has been integral for optimizing performance. LLMs, such as GPT-3 (Brown et al., 2020), ChatGPT-4 (OpenAI, 2023), Llama-1 (Touvron et al., 2023a), and Llama-2 (Touvron et al., 2023b), are exemplars of this trend, showcasing enhanced capabilities when furnished with explicit human-generated instructions. Conventionally, this entailed considerable human input in both instruction creation and response generation, leading to expansive datasets for fine-tuning (Stiennon et al., 2020; Ouyang et al., 2022).

Emerging from predominantly human-instructed models, a **crucial inquiry** emerges: Can LLMs be fine-tuned to adeptly handle complex instructions *without* human feedback? The potential gains from this direction are multifaceted. Chief among them is the direct cut in costs tied to human annotations (Askell et al., 2021). Beyond the monetary aspect, such a transition also alleviates potential biases seeded by human curators (Gallegos et al., 2023). A loftier, albeit less conspicuous aim, is to amplify the quality of instructions and boost model performance. Several methodologies have surfaced in pursuit of these merits.

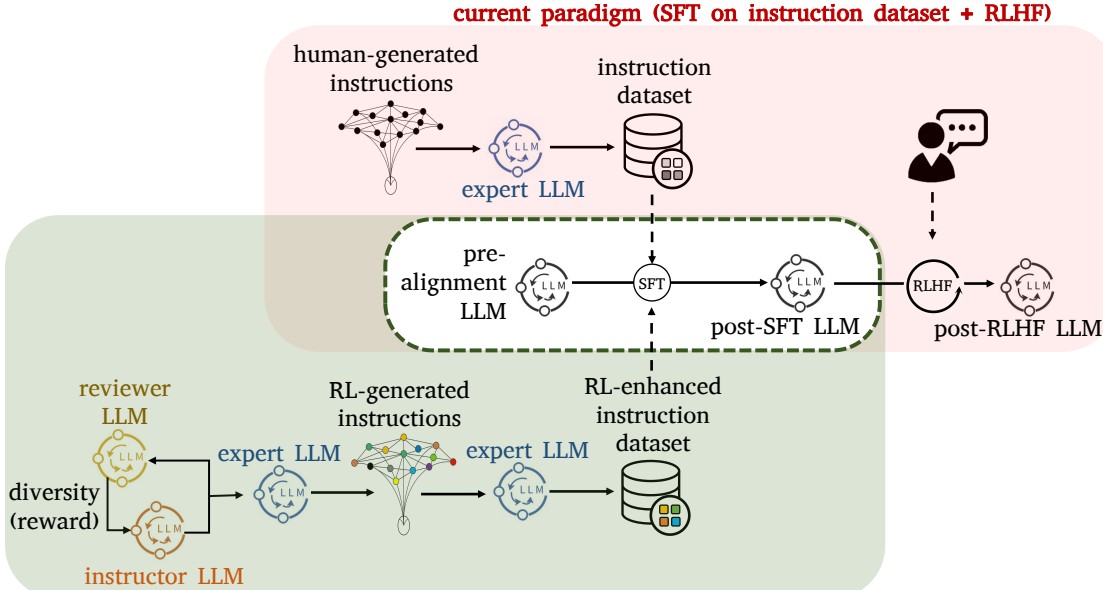

Figure 1: **Comparative overview of LLM alignment techniques.** Current methods (red shaded region) typically involve a two-phase process, starting with Supervised Fine-Tuning (SFT) of a pre-aligned LLM using a dataset of human-crafted instructions and corresponding responses (often sourced from an *expert LLM* like ChatGPT), leading to a post-SFT LLM. This is then fine-tuned using RLHF, where human feedback on preferences is incorporated, resulting in a post-RLHF LLM. In contrast, our TeaMs-RL method (green shaded region) employs a single-phase SFT approach, initially utilizing RL for teaching expert LLMs to generate high-quality instructions. We train an RL policy (*instructor LLM*) to create diverse instructions (with the diversity evaluated by a *reviewer LLM* as a reward signal). Once trained, the instructor LLM produces a set of actions to teach an *expert LLM* to generate high-quality instructions, and the instructions are leveraged to query the expert LLM to form the SFT instruction dataset. This approach capitalizes on the strengths of RL to enhance the complexity of instructions and consequently the value of responses from the *expert LLM*. Note that the expert LLMs are not involved in the training of the RL policy; we only use the instructor and the reviewer LLM for training the RL policy. The expert LLM is used to generate instructions and corresponding responses under the guidance of the trained RL policy.

The self-instruct method by Wang et al. (2022) pioneers this direction by using some external LLM to generate responses to human-generated seed instructions for instruction dataset curation. A notable recent development is the evolutionary strategy presented by Xu et al. (2024). Here, LLMs are seeded with initial instructions, gradually evolving towards generating more complex directives within predefined constraints. Despite their method's commendable performance relative to alternative models, it necessitates a multitude of interactions with an *expert LLM* (e.g., advanced LLMs such as ChatGPT), potentially raising concerns regarding resource demands.

In this research, we propose a novel method to improve instruction quality with the principles of Reinforcement Learning (RL) (Sutton & Barto, 2018), hence enhancing LLMs' ability to comprehend and effectively execute intricate instructions without human involvement, while also boosting model privacy protection. Our method, outlined in the green shaded region of Fig. 1, begins with training an instructor LLM (RL policy) for teaching an expert to generate diverse, complex instructions. These are then used to elicit expert LLM responses, forming a diversity-enhanced instruction dataset. The *final* stage of supervised fine-tuning (SFT) of a pre-alignment LLM (e.g., Llama-1) with this dataset enhances its complex task processing capabilities. Notably, our study shows that the *thoughtful framing of instructions (using RL) is equally, if not more, crucial than generating responses from external sources such as expert LLMs or human feedback.* Moreover, to clarify our methodology and facilitate replication, we provide a detailed example illustrating the inputs and outputs at each stage of our process, similar to Figure 1, see Appendix A.

The key benefits of our methodology include: Firstly, a reduction in the need for human instructors (e.g., annotators and evaluators), offering a more cost-effective alignment paradigm that supports the continuous development of capable and affordable LLMs. Secondly, our approach moderates the need for frequent queries to external models, yielding monetary benefits and mitigating the environmental impact of already power-hungry data centers (Dhar, 2020; Wu et al., 2022). In the same experimental settings, our experiment results demonstrate that our method shows better performance than strong baselines such as WizardLM, while significantly **reducing the number of expert LLM queries by over 94%**. This lesser dependence on external data (for either SFT or RLHF) can potentially broadens the scope to other engineering and science disciplines where data availability is limited or data collection is costly. This aspect is also crucial in mitigating privacy concerns (Dong et al., 2022; Ko et al., 2023).

## 2 Related Work and Contribution

A multitude of studies have explored the training of language models using instructions paired with their respective responses. Notable works in this realm include GPT-3 (Brown et al., 2020), ChatGPT-4 (OpenAI, 2023), Flan collection (Longpre et al., 2023), Flan models (Wei et al., 2021), and Alpaca (Taori et al., 2023). The prevalent methodology often requires human annotators to craft instructions and curate corresponding responses, leading to the assembly of detailed instruction-response datasets. Such datasets, exemplified by the one used in GPT-3's training (Brown et al., 2020), tend to resonate well with human preferences and markedly improve language model proficiency. However, (crowd) sourcing these datasets through human means can be not only costly but also prone to issues of bias (Gallegos et al., 2023).

Conversely, the self-instruct approach (Wang et al., 2022) charts a different course by tapping into the potential of LLMs to autonomously generate both instructions and their responses, facilitated by the provision of predefined seeds. While this strategy alleviates the dependence on human effort, it might not consistently capture the breadth and depth of diverse instructions and responses typically achieved with human annotators. Building on the pioneering self-instruct methodology, Xu et al. (2024) introduced WizardLM, an evolutionary instruction approach. In WizardLM, initial instructions drawn from the "Evol Alpaca" dataset undergo adaptation through the amalgamation of command instructions and expert LLMs such as ChatGPT. Owing to its commendable performance in both formulating and adhering to intricate instructions, WizardLM has garnered considerable attention. However, its reliance on *random sampling* of command instructions could fall short in improving the breadth and richness of instructions fed to the LLMs. Furthermore, the heavy dependence on external expert models—commonplace in prevailing self-instruct methods—poses concerns not just economically, but also in terms of environmental impact.

Departing from these precedents, our methodology uniquely uses RL—to help generate foundational fine-tuning instruction data rather than for post-tuning refinements (such as RLHF (Stiennon et al., 2020; Ouyang et al., 2022)). Our key **technical contribution** is the formulation of a Markov Decision Process (MDP) to train an instructor LLM as an RL policy, adeptly tailored for contextualized instruction manipulations, thereby enabling alignment through SFT on small, high-quality datasets. In contrast to WizardLM's approach of treating text manipulations as *ordinal* selections, our MDP framework encodes them in a continuous action space, enabling a more refined differentiation of the intricate effects of textual manipulations (carried out by the instructor LLM) on the quality of instructions. Also, unlike the intricate task of crafting well-structured tree instructions required by Tree-Instruct (Zhao et al., 2023b), our MDP can be resolved using TRPO (Schulman et al., 2015) (with other common methods being applicable). This approach alleviates the combinatorial complexity inherent in sequential instruction actions, allowing for iterative policy refinement (with TRPO offering the added benefit of guaranteed monotonic progression). Empirically, we demonstrate the efficacy of SFT alignment on a small but high-quality dataset across benchmarks such as ARC and HellaSwag with foundational pre-aligned models such as Llama-1 and Llama-2 (refer to Section 4 for details). Additionally, comparative experiments on model privacy attacks reveal that our method markedly enhances model privacy protection (Section 4.3).

# 3 Method

We first train an instructor LLM policy (Sec. 3.1) based on a continuous action space encoding (Sec. 3.1.1) and diversity rules as a reward function (Sec. 3.1.2), a strategy designed to foster the generation of high-quality instructions with an expert LLM such as ChatGPT. Subsequently, using this policy and the expert LLM, we create an instruction-response dataset to fine-tune a pre-aligned LLM. Importantly, our enriched dataset allows bypassing the usual RLHF phase, resulting in a post-SFT LLM that is already proficient in responding to complex instructions. Also, the instruction policy is transferable for aligning various foundation LLMs such as Llama-1-chat-7b and Llama-2-chat-7b. Note that our method has been primarily tested on Llama models. There may be performance variations when applying our method to other models. This includes considerations regarding differences in model architecture and computational requirements, which could impact the effectiveness of our approach when transferred to models beyond Llama.

## 3.1 Training the Instructor LLM

To train the RL policy (i.e., instructor LLM[1]), we utilize the open-source WizardLM-13b model[2,3] as the reviewer LLM, which provides a diversity score for the current instruction set as the reward signal. WizardLM-13b, being a cost-effective option, can capture instruction nuances for reliable reward evaluation (Sec. 3.1.2).

The training procedure comprises several key stages. First, we select a single initial instruction, such as "How to cook food." This chosen instruction is then input into our instructor model. Second, we leverage the reviewer LLM to evaluate the instructor model's performance and provide reward signals. The policy training concludes if the rewards demonstrate convergence or the iteration limit is reached. Following this iterative training phase, we use the instructor LLM to generate complex instructions using a tailored action space (Sec. 3.1.1). In the final phase, the reviewer LLM assesses the diversity of the generated instructions to ensure instruction quality (Sec. 3.1.2).

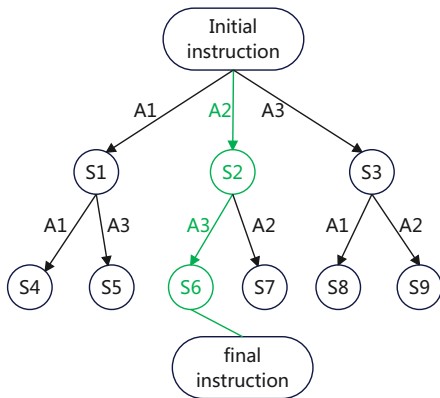

Figure 2: RL policy search for LLM instruction generation. A denotes actions (prompts), S denotes states (instructions), and the green line indicates the optimal instruction generation achieved during the policy search.

To further illustrate how to leverage RL to train the instructor, we first model the RL process with a Markov Decision Process (MDP). An intuitive example is shown in Figure 2, it illustrates RL policy search for LLM instruction generation, where A represents actions (prompts), S represents states (instructions), and the green path denotes the optimal instruction generation. An MDP is present as $(\mathcal{S}, \mathcal{A}, P, r, \gamma)$, where $\mathcal{S}$ denotes the state space, $\mathcal{A}$ denotes the action space, the transition probability function is $P : \mathcal{S} \times \mathcal{A} \times \mathcal{S} \to [0, 1]$, the reward function is denoted as $r : \mathcal{S} \times \mathcal{A} \to \mathbb{R}$ and $\gamma$ denotes the discount factor. In an MDP, we aim to maximize the expected cumulative reward values, which are defined as $Q^\pi(s, a) = \mathbb{E}\left[\sum_{t=0}^{\infty} \gamma^t r\left(s_t, a_t\right) \middle| \pi, s_0 = s, a_0 = a\right]$ and $V^\pi(s) = \mathbb{E}\left[\sum_{t=0}^{\infty} \gamma^t r\left(s_t, a_t\right) \middle| \pi, s_0 = s\right]$ for state-action pairs and states, respectively. Moreover, we have the advantage function $A^\pi(s, a) = Q^\pi(s, a) - V^\pi(s)$. Thus, the learning problem is formulated as Equation (1), where $\rho$ denotes an initial state distribution.

$$\max_{\pi \in \Pi} V^\pi(\rho) \coloneqq \mathbb{E}_{s \sim \rho}\left[V^\pi(s)\right]. \tag{1}$$

In our framework, reward values are assigned by reviewer LLMs, which assess the diversity of actions within a trajectory. A trajectory with highly diverse actions receives a higher reward. The state $s_t \in \mathcal{S}$ on a trajectory $\mathcal{T}$ represents the generative instructions. Action space $\mathcal{A}$ is designed in Sec. 3.1.1, and specific

---

[1]The instructor LLM consists of an RL network and a WizardLM-13b model.
[2]https://github.com/nlpxucan/WizardLM
[3]https://huggingface.co/WizardLM/WizardLM-13B-V1.2

reward settings are provided in Sec. 3.1.2. On the basis of these settings, we can employ a model-free RL algorithm (e.g., TRPO Schulman et al. (2015)) to train the instructor LLM.

### 3.1.1 Action Set

Inspired by WizardLM (Xu et al., 2024), we leverage several actions to generate complex instructions, including *"breadth action", "add constraint", "deepening", "concretizing", "increase reasoning steps", "complicate input"* (Xu et al., 2024). However, distinct from WizardLM, we map this discrete set into a continuous space by using a language encoding to represent each action, which is then used for a policy to generate instructions. This enables inherently capturing contextual nuances for direct comparison between actions via their Q values. The details of each action are in Appendix C.

### 3.1.2 Reward Settings

In this section, consider an evaluation prompt $g$. If $g$ is characterized as "equal", the corresponding reward $r$ is designated as 0. Otherwise, $r$ is set to 1. The term "equal" here refers to semantic equivalence. It is noteworthy that the judgment is determined by the reviewer LLM. For instance, when conformity exists between the initial and subsequent instructions, the reward $r$ is 0. In contrast, a lack of conformity yields a reward value of 1.

For example, an *evaluation prompt = """ Here are two Instructions to ChatGPT AI, do you think they are equal to each other, which meet the following requirements: 1. They have same constraints and requirments. 2. They have same depth and breadth of the inquiry. The First Instruction: """ +instruction + """. The Second Instruction: """ + state + """. Your Judgement (Must Just only answer: Equal or Not Equal. No need to explain the reason; 'Equal' and 'Not Equal' are not allowed to appear simultaneously.): """*.

### 3.2 Fine-tune a Pre-Aligned LLM

As illustrated in Fig. 1, an approach is adopted to optimize cost-efficiency considerations during the phase dedicated to fine-tuning a pre-aligned LLM. Specifically, we use the initial instructions from the Alpaca dataset.

The fine-tuning process is executed as follows: We first employ the Alpaca dataset[4] as the initial set of instructions and input the initial instructions into the expert LLM like ChatGPT. Next, we apply our trained instruction generation policy (derived from the Instructor LLM) to guide the Expert LLM (ChatGPT) in generating high-quality instructions. Specifically, this policy acts as a teacher directing the Expert LLM to produce instructions that meet our quality criteria. The instructor LLM provides different responses during the RL training process. Expert LLM is used to generate instructions and their associated responses, it serves as the source of knowledge and expertise. Then, the expert LLM produces responses to these high-quality instructions. With the instructions and responses in hand, we fine-tune the pre-aligned LLMs like Llama-1-chat-7b and Llama-2-chat-7b via a supervised fine-tuning way (SFT)—a cost-effective way to enhance LLM capabilities. Finally, we can have the post-SFT LLMs, such as TeaMs-RL-1-7b.

---

**Algorithm 1** The Pipeline of training LLMs.

---

1: Design a set $A$ of actions and reward settings.
2: With reviewer LLMs' evaluation, leverage the principle of RL and an instructor LLM to search for a policy $\pi$.
3: Utilize the trained policy to teach expert LLMs to generate high-quality instructions and corresponding responses.
4: Fine-tune a pre-aligned LLM with the generated instructions and corresponding responses.

---

[4]https://huggingface.co/datasets/tatsu-lab/alpaca

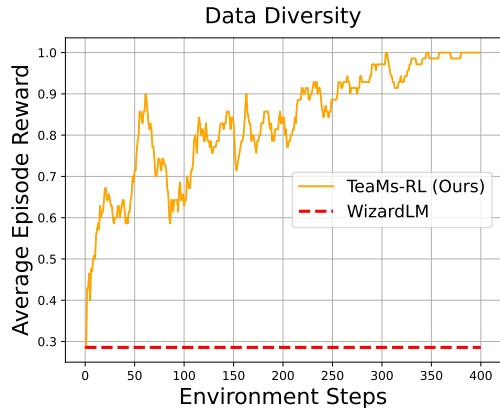

Figure 3: Compare our method with WizardLM in terms of data diversity.

### 3.3 Practical Algorithms

The practical training process is outlined in Algorithm 1, which describes the training pipeline. A more detailed algorithm can be found in Algorithm 2, provided in Appendix B. In our training process, in order to teach expert LLMs how to generate high-quality instructions, we leverage the RL policy we have trained to prompt expert LLMs to generate instructions step by step. Building on the training process, we fine-tuned an LLM called "TeaMs-RL". Note that we have the flexibility to choose any other RL methods for policy training, here we choose TRPO (Schulman et al., 2015) (TRPO is not the only choice), though it has longer training times versus alternatives like Proximal Policy Optimization. We select TRPO for its rigorous advantage function handling and theoretical guarantees of monotonic improvement.

## 4 Experiments

### 4.1 Enhancement of Instruction Diversity

We deploy a policy that is designed to orchestrate a trajectory consisting of six distinct instruction actions, in accordance with the training process to facilitate the generation of instructions by expert LLMs. *Note, the expert LLMs utilized in our study, as well as those used by WizardLM, are ChatGPT-3.5 and ChatGPT-4. Both models were accessed in 2023, ensuring that we employed the same versions of the ChatGPT-3.5 and ChatGPT-4 models to generate instructions and responses.* A noteworthy aspect of our investigation involves a comparative assessment of the data quality resulting from the utilization of our policy, in contrast to the approach adopted by WizardLM (Xu et al., 2024), which relies upon random sampling for data generation by querying expert LLMs. It bears emphasizing that our instruction actions are randomly initialized, but as training progresses, our policy iteratively learns to enable the instructor LLM to produce increasingly complex and diverse instructions.

As shown in Fig. 3, our method reliably enhances the diversity score of the instruction set.[5] The main computation overhead is learning the instruction policy on the relatively small WizardLM-13b in less than 1 hour on 2 NVIDIA RTX A6000 GPUs (with 896 total queries). This results in a transferable policy that reduces alignment cost across models compared to RLHF's per-model RL. Offloading to policy learning provides an instruction set for joint tuning and alignment—a substantial benefit over tuning-only data usage. For detailed information on how to teach expert LLMs to generate instructions, please refer to Appendix D.

---

[5]In designing instruction actions, the "breadth action" is a single action regarding breath thoughts. Thus, we insert the action into the middle of our trajectory to enhance breadth instructions after training a policy.

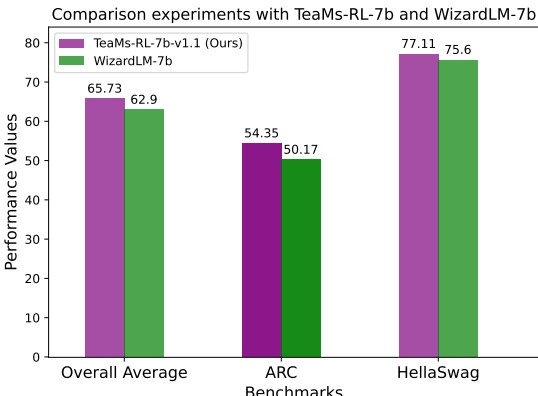

Figure 4: Compare our method with WizardLM 7B on LM-Eval Benchmarks (the higher the value, the better the method's performance).

## 4.2 Compare with WizardLM-7b on ARC and HellaSwag Benchmakrs

To comprehensively examine the effectiveness of our method, we carry out experiments on popular benchmarks: (1) AI2 Reasoning Challenge (ARC) benchmark (Clark et al., 2018): The benchmark introduces a fresh question set and baselines, all strategically curated to foster AI research in the realm of advanced question answering, setting a significantly higher bar for knowledge and reasoning capabilities compared to previous challenges. (2) HellaSwag benchmark (Zellers et al., 2019): The benchmark introduces a challenging dataset, revealing that even state-of-the-art models struggle with commonsense inference, as evidenced by the significant performance gap between humans (95% accuracy) and models (48%), achieved through adversarial filtering, a robust data collection paradigm that selects adversarial machine-generated wrong answers by scaling up the length and complexity of dataset examples to a 'Goldilocks' zone where the text generated is absurd to humans yet often misclassified by models.

We trained a llama-1-7b model denoted "TeaMs-RL-7b-v1.1" with our dataset of 17,878 instruction-response pairs, the training time is about 2 hours on 4 NVIDIA RTX A6000 GPUs. As shown in Figures 4 and 5, our method shows superior performance over WizardLM-7b models[6] in the same experimental settings. It is crucial to highlight an important aspect of our methodology in relation to data utilization. The dataset employed for training our model is approximately **one-fourteenth** the size of the dataset utilized by WizardLM, as illustrated in Figure 5 (a). Furthermore, Fig. 5 (b) highlights the discernible difference in the query count posed to ChatGPT models between our method and WizardLM, with the latter requesting ChatGPT for responses more than **seventeen times** than TeaMs-RL. This marked contrast underscores the **cost-effectiveness of our data generation approach**, which mitigates the expenses associated with dataset acquisition and generates **high-quality data**. It highlights that our method is a more economically viable and sustainable strategy for training LLMs. For more experimental results, please see Appendix G. Note that the ChatGPT models we used to query data show performance equivalent to the ChatGPT models used in the baselines.

In our comparison experiments, we take the same settings: 25 shots for ARC, 10 shots for HellaSwag. In our experimental setup, all models are configured with a float16 format. We compare to WizardLM-7b since both this approach and ours use llama-1-7b as the base model. Notably, WizardLM-7B queried ChatGPT 624,000 times for responses, whereas our method queried open-source WizardLM13B 896 times during policy training and ChatGPT 35,756 times during high-quality data generation; our total queries are substantially fewer. Therefore, we believe the comparison is fair in terms of matched base model and vastly lower query amount. Even with the query count of WizardLM-13B, $35756 + 896 = 36652$, our method can still reduce the query count by 94.13% to reach comparable outcomes, emphasizing a more economical and sustainable strategy for LLM training.

---

[6] https://huggingface.co/TheBloke/wizardLM-7B-HF

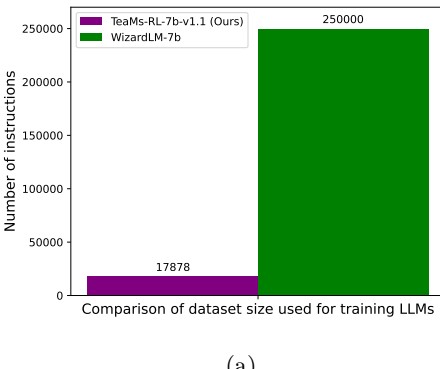 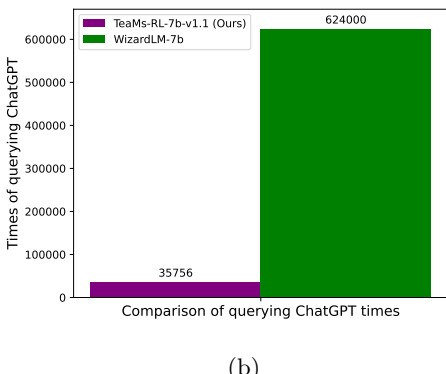

(a)                                                    (b)

Figure 5: Compare with WizardLM 7B on dataset size used for training LLMs (a) and querying count of advanced LLMs (b) (the lower the value, the better the method's performance).

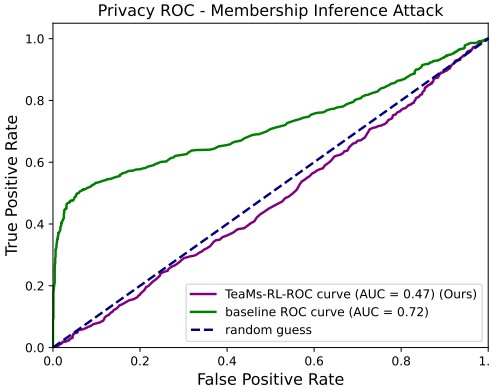

Figure 6: Privacy Attacks on the Model: Our model demonstrates strong privacy protection performance. The more closely the ROC curve of the model aligns with random guessing, and the closer the AUC value of the model approaches 0.5, the stronger the indication of improved privacy protection by the model.

### 4.3 Experiments of Model Privacy Attack

AI safety has increasingly adopted data synthesizers designed to produce differentially private datasets to mitigate the risk of inadvertent data leakage (Dong et al., 2022). These datasets serve as the foundational element for training machine learning algorithms. However, it presents a dilemma wherein practitioners have to choose between large training data and data privacy. Addressing this problem, our method tries to handle the dilemma of large training data and data privacy. It aims to enhance the performance of training models while ensuring improved data privacy, even with limited data.

As illustrated in Fig. 6, we have conducted a series of membership inference attack (Shokri et al., 2017; Carlini et al., 2022) experiments to assess our model's privacy performance rigorously. Our model exhibits a Receiver Operating Characteristic (ROC) curve that closely approximates random guessing, yielding an Area Under the Curve (AUC) value of 0.47 in this evaluation. Conversely, the baseline model, trained on a dataset comprising 44,000 samples, displays a notable deviation from random guessing, with an AUC value of 0.72.

In model privacy assessment, a closer proximity of the ROC curve to the random guess curve indicates better model privacy protection performance, while an AUC value approaching 0.5 further suggests better model

privacy protection (Ye et al., 2022). The experiment results demonstrate the substantial enhancement in model privacy protection performance achieved by our model relative to the baseline model.

### 4.4 Comparison Experiments on Apacal Eval Benchmarks

To comprehensively evaluate the effectiveness of our method, we carried out additional experiments using the apacal eval benchmark[7], a comprehensive tool for evaluating model performance across a diverse range of tasks. As shown in Table 1, the results of these experiments demonstrate that our model attains a win rate of 13%. In comparison, the baseline model, WizardLM-7b Xu et al. (2024), achieves a win rate of only 4.84%. The experimental results further confirm that our method surpasses the performance of WizardLM Xu et al. (2024). For the details of the apacal eval Leaderboard, please see the Link: `https://tatsu-lab.github.io/alpaca_eval/`.

| Method \ Task | apacal eval benchmark |
|---|---|
| TeaMs-RL-1-7b (ours) | **13%** |
| WizardLM-7b | 4.84% |

Table 1: Comprehensive evaluation experiments on the apacal eval benchmark: Our model achieved a win rate of 13%, significantly higher than the baseline model, WizardLM-7b, which recorded a win rate of 4.84%.

### 4.5 Comparison Experiments of Solving General Tasks

In this evaluation, we present scenarios where random tasks are provided, and the subsequent response is generated by our model (TeaMs-RL-1-7b) and baseline models. Notably, our model performs exceedingly better than the baseline models in providing a well-structured and accurate response to the given instruction. More cases are provided in Appendix H and Appendix I.

> Task: "Please suggest two examples of animals that fly."

> TeaMs-RL-1-7b output: "Certainly, here are two examples of animals that fly:
> 1. **Birds**: Birds are one of the most well-known examples of animals that fly. They have evolved wings that are lightweight, flexible, and covered with feathers. Birds can fly by flapping their wings or gliding through the air using air currents. They can also soar in the air without flapping their wings, which is called sustained flight. Some birds, like the albatross, can even fly for days without stopping.
> 2. **Insects**: Insects, such as bees, wasps, and moths, are also capable of flight. They have wings that are lightweight and covered with a thin layer of skin called cuticle. Insects fly by beating their wings, which creates lift and propels them through the air. Some insects, like moths, can also hover in the air by adjusting the angle of their wings.
> These are just two examples of animals that fly. There are many more, such as bats, pterosaurs, and even some species of fish that can glide through the water using their fins."

> llama-1-7b output: "I'm sorry, I don't understand the question."

> WizardLM-7b output: "Certainly! Two examples of animals that fly are birds and bats. Birds are known for their ability to fly using wings, while bats use wings to glide through the air. Both are amazing creatures that have evolved to navigate the skies. Is there anything else I can help you with?"

---

[7]`https://github.com/tatsu-lab/alpaca_eval`

### 4.6 Comparison Experiments of Solving a Math Problem

We conduct comparison experiments with several strong baseline models on mathematical tasks. The experimental procedure entails inputting instructions into LLMs, with the LLMs subsequently generating responses aligned with the provided instructions. The empirical findings from our study reveal that the post-alignment LLM, TeaMs-RL-7b, exhibits superior mathematical capabilities compared to the baseline models. This enhanced performance is noteworthy, particularly when it surpasses even those models with a larger parameter count, such as the 13b models. This outcome underscores the efficacy of our alignment and training methodologies in augmenting the specific skill set of LLMs, particularly in mathematical problem-solving. More results are introduced in Appendix J.

---

Task: "Given that $f(x) = 5x^3 - 2x + 3$, find the value of f(2)."

---

TeaMs-RL-7b output: "Hello there! I'm happy to help you with your question. To find the value of f(2), we need to plug 2 into the function $f(x) = 5x^3 - 2x + 3$.
So, $f(2) = 5(2)^3 - 2(2) + 3 = 5(8) - 2(2) + 3 = 40 - 4 + 3 = 39$.
Therefore, the value of f(2) is 39. I hope this helps! Let me know if you have any other questions."
The output is correct.

---

WizardLM-7b output: "Sure, I can help you with that!
To find the value of f(2), we simply need to substitute x = 2 into the function and simplify:
$f(2) = 5(2)^3 - 2(2) + 3$
f(2) = 30 - 4 + 3
f(2) = 21
Therefore, the value of f(2) is 21.
Is there anything else I can help you with?"
The output is wrong.

---

Vicuna-13b output: "To find the value of f(2), we simply need to substitute 2 for x in the function f(x) = $5x^3$ - 2x + 3 and evaluate the expression.
So, f(2) = $5(2)^3$ - 2(2) + 3= 5(4) - 2(2) + 3= 20 - 4 + 3= 13
Therefore, the value of f(2) is 13."
The output is wrong.

---

Llama-2-chat-13b output: "Hello! I'd be happy to help you with that! To find the value of f(2), we need to plug 2 into the function f(x) = $5x^3$ - 2x + 3.
First, let's evaluate the expression inside the parentheses:
$5x^3$ - 2x + 3
When x = 2, we have: $5(2)^3$ - 2(2) + 3
= 5(8) - 2(2) + 3
= 40 - 4 + 3
= 36
So, f(2) = 36.
Is there anything else you'd like to know?"
The output is wrong.

---

ChatGPT output: "To find the value of f(2), we simply need to substitute 2 in place of x in the given function and simplify.
$f(2) = 5(2)^3 - 2(2) + 3$
f(2) = 5(8) - 4 + 3
f(2) = 40 - 1
f(2) = 39
Therefore, the value of f(2) is 39."
The output is correct.

### 4.7 Compared to Llama-2-7b-chat Models

As depicted in Table 2, we conduct a comparative analysis of our model against the llama-2-chat-7b model, which is trained using RLHF. The experimental findings indicate that our model, which is developed on the llama-1-7b framework, surpasses the llama-2-chat-7b in performance across both the ARC and Hellaswag benchmarks. Specifically, our model achieved a score of 78.35 on the Hellaswag benchmark and 55.89 on the ARC benchmark, which are markedly higher than the scores of 77.74 and 53.07, respectively, recorded by the llama-2-chat-7b on these benchmarks. This analysis underscores the superior efficacy of our approach.

| Method \ Task | ARC | Hellaswag |
|---|---|---|
| TeaMs-RL-1-7b (ours) | **55.89** | **78.35** |
| llama-2-chat-7b | 53.07 | 77.74 |

Table 2: A comparison experiment regarding our model and a llama-2-chat-7b model: our model outperforms the llama-2-chat-7b on the ARC and Hellaswag benchmarks, scoring 78.35 and 55.89, respectively, versus 77.74 and 53.07 achieved by the llama-2-chat-7b.

### 4.8 Ablation Experiments Regarding Data size

To further evaluate the effectiveness of our proposed method, we conduct ablation experiments to analyze the impact of dataset size, as summarized in Table 3. Expanding the dataset from 17,878 to 19,395 entries led to notable performance improvements for the TeaMs-RL-7b model on both the Hellaswag and ARC Challenge benchmarks. Specifically, the model's performance on the Hellaswag benchmark improves from 77.17 to 78.35, while on the ARC Challenge benchmark, it increases from 54.35 to 55.89. Further expanding the dataset from 19,395 to 21,396 entries yields more improvements: the Hellaswag benchmark score increases from 78.35 to 78.59, and the ARC Challenge score improves from 55.89 to 56.74. These results provide further confirmation of our method's effectiveness. It is worth noting that the performance improvements are not strictly linear with the increase in dataset size; however, adding more data generally proves beneficial for enhancing model performance.

| Data Size \ Task | ARC | Hellaswag |
|---|---|---|
| 17,878 | 54.35 | 77.17 |
| 19,395 | 55.89 | 78.35 |
| 21,396 | 56.74 | 78.59 |

Table 3: Ablation experiments regarding the impact of data size: Expanding the dataset from 17,878 to 21,396 entries improves the performance of the TeaMs-RL-7b model on both the Hellaswag and ARC Challenge benchmarks.

## 5 Conclusion

In this study, we depart from the traditional RLHF paradigm and introduce a method that amplifies instruction quality while significantly cutting the costs linked to querying proprietary LLMs such as ChatGPT, a

feat realized through strategic RL application in autonomous instruction set generation. Leveraging this methodology, we teach LLMs to generate **high-quality data**, refine a foundational model, and conduct comprehensive experiments to assess its efficacy. Remarkably, the LLM trained under this framework rivals the performance of the acclaimed WizardLM, despite being constrained by a significantly smaller dataset and fewer query instances: our dataset amounts to a mere 6.75% of WizardLM's, and the query counts to ChatGPT are only 5.73% of what WizardLM uses. This highlights the economic and sustainable advantages of our approach, underscoring its potential to enhance data quality within budgetary confines.

Moreover, our method can improve model performance while effectively mitigating model privacy leakage risks. Our experiments clearly demonstrate substantially enhanced privacy protection over the baseline model.

Beyond mere practical implications, our findings question the conventional two-stage LLM training pipeline, suggesting that perhaps it is possible to train proficient LLMs without the necessity of human feedback. It beckons a rethinking of the pivotal role humans play in LLM training, urging a more judicious deployment of human resources to truly critical facets of the training process.

## 6  Limitations

Our method leverages RL to teach LLMs how to generate high-quality instruction fine-tuning data, aiming to diminish collection costs and the dependency on human alignment; however, it is important to note that this research did not explore the potential benefits of incorporating an additional stage of human feedback, which could potentially enhance alignment. This leaves open the question of whether data derived from expert LLMs sufficiently aligns with human values or inherits human value deficiencies, such as biases. While our approach successfully reduces the reliance on external models like expert LLMs, it does not entirely eliminate their use. The full extent to which these models can be excluded remains an area for future exploration. Hence, while our work paves the way for greater autonomy and efficiency in data alignment, it simultaneously raises more questions than it answers, necessitating additional exploration.

Moreover, the policy employed in instructing expert LLMs is trained using specific instructions, e.g., "How to cook food". This policy is then applied to guide expert LLMs in generating instructions from various other initial inputs. However, it is important to note that this policy may not be universally precise for all initial instructions. Training a distinct policy for each initial instruction presents a significant challenge, as it could be both time-consuming and costly.

In the future, our research will delve into the complex interplay between these policies' performance and their training's associated costs. Additionally, we aim to explore the alignment of LLMs with human values, examining how these aspects can be balanced and optimized. This investigation is crucial for enhancing the efficiency and applicability of LLM training, ensuring both practicality and alignment with ethical standards.

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

# A  A Methodological Example

To clarify our methodology and facilitate replication, we provide a detailed example illustrating the inputs and outputs at each stage of our process, similar to Figure 1 in our paper.

**Stage 1: Policy Training**

**Components:**

- **RL Policy:** (derived from Instructor Model) Learns to select actions based on the reviewer's feedback to improve the instructions.

- **Reviewer Model:** Provides feedback on the instructions.

**Example:**

1. **RL Policy Training**
   With any Initial Instructions such as:

   > "Describe the process of photosynthesis."

   The RL policy with Instructor Models learns to select actions that diversify instructions based on feedback from the reviewer model.

2. **RL Policy Training:**
   Based on the reviewer's feedback, the RL policy learns to select actions such as:

   - *Add Constraints*
   - *Deep Reasoning*
   - *Width Reasoning*

**Stage 2: RL Policy Action Selection**

**Selected Action:** "Add Constraints"

**Purpose:** To make the instruction more challenging and comprehensive by adding specific constraints or requirements.

**Stage 3: Guiding Expert LLMs**

**Purpose:** To make the instruction more challenging and comprehensive by adding specific constraints or requirements.

RL Policy Action Selection such as: *"Add Constraints"*

The RL policy generates a specialized prompt based on this action to guide the expert LLM in rewriting the instruction.

**Add Constraints: Action Prompt to Expert LLM:**

> """ I want you to act as a Prompt Rewriter.
> Your objective is to rewrite the given prompt into a more complex version to make it more challenging for AI systems like ChatGPT and GPT-4.
> Ensure that the rewritten prompt remains reasonable, understandable, and suitable for human response.
> Do not omit any non-text parts such as tables or code in the given prompt.
> Do not repeat conditions or requirements in your response, and do not disclose your role.
> Provide only the rewritten prompt without any introduction or explanation.
> The new prompt should not exceed 2048 words.
> You should complicate the given prompt by adding one more constraint or requirement.
> Try not to make the rewritten prompt verbose; you can only add or replace 10 to 20 words in the given prompt.
> Do not include phrases like 'Given Prompt' or 'Rewritten Prompt' in your response.
> Given Prompt: "Describe the process of photosynthesis."

**Stage 4: Expert LLM Generates Rewritten Instruction**

**Expert LLM Output:**

> "Describe the process of photosynthesis in plants and explain how it varies in different environmental conditions."

**Explanation:**

- **Original Instruction:** Simple and straightforward.

- **Rewritten Instruction:** Adds the constraint of explaining variations in different environmental conditions, increasing complexity and depth.

- **Compliance with Guidelines:** The expert LLM added approximately 12 words, adhering to the limit of adding or replacing 10 to 20 words.

Once we have the generation instructions, we use them to query the expert model for corresponding responses. After obtaining the responses, we have the final dataset (instructions and corresponding responses), which is then used to fine-tune foundation models.

**Summary**

This example demonstrates how our RL policy guides expert LLMs to generate more complex and high-quality instructions by selecting appropriate actions. The process ensures that the generated instructions are challenging yet reasonable, facilitating the creation of a valuable dataset for training advanced AI models.

# B  Practical Algorithm

Here, we introduce Trust Region Policy Optimization (TRPO) (Schulman et al., 2015), which is outlined in Algorithm 3. In this context, $\rho_\pi$ represents the discounted state visitation frequencies, while $\epsilon = \max_{s,a} |A_\pi(s,a)|$ denotes the maximum absolute advantage function value across all states and actions. The performance objective, $\eta(\pi)$, is defined as the expected cumulative reward over time, $\eta(\pi) = \mathbb{E}_{s_0,a_0,\dots} [\sum_{t=0}^{\infty} \gamma^t r(s_t)]$, where $\gamma$ is the discount factor.

---

**Algorithm 2 TeaMs-RL:** Teaching LLMs to Generate Better Instruction Datasets via RL.

---

1: Initial dataset $D_1$, instruction dataset $D_2$, response dataset $D_3$, time step $t$, parameters $\theta$, the number of instructions and responses $n$, RL batch size number $x$.
2: With reviewer LLMs' evaluation, leverage the principle of RL and an instructor LLM to search for a policy $\pi$.
3: **for** $t \leftarrow 1$ **to** $T$ **do**
4:     Collect trajectories $D_t = \{\tau_x\}$ with policy $\pi_t$ in the reviewer LLM interactions.
5:     Compute advantage $A_t$ and estimate gradient

$$g_k = \frac{1}{|D_t|} \sum_{\tau \in D_t} \sum_{t=0}^{\tau} \nabla_\theta \log \pi(a_t|s_t, \theta_t) A_t.$$

6:     Use the methods of conjugate gradient and line search to search the policy parameters $\theta_{k+1}$ via TRPO.
7: **end for**
8: Receive the trained policy $\pi\prime$ that is parameterized by $\theta_T$
9: **for** $i \leftarrow 1$ **to** $n$ **do**
10:     Leverage the trained policy $\pi\prime$ to teach expert LLMs to generate high-quality instruction $i$, which can be used to build dataset $D_2$ with the initial dataset $D_1$.
11: **end for**
12: **for** $j \leftarrow 1$ **to** $n$ **do**
13:     Utilize expert LLMs to query response $j$ for the generated instruction $i$, which can be used to build dataset $D_3$ with the initial dataset $D_2$.
14: **end for**
15: Fine-tune a pre-aligned LLM with the generated instructions and corresponding responses.

---

**Algorithm 3** TRPO(Schulman et al., 2015)

---

Initialize $\pi_0$.
  **for** $i = 0, 1, 2, \ldots$ until convergence **do**
    Compute all advantage values $A_{\pi_i}$.
    Solve the constrained optimization problem

$$\pi_{i+1} = \arg\max_\pi \left[ L_{\pi_i}(\pi) - C D_{\mathrm{KL}}^{\max}(\pi_i, \pi) \right]$$

    where $C = \frac{4\epsilon\gamma}{(1-\gamma)^2}$ and $L_{\pi_i}(\pi) = \eta(\pi_i) + \sum_s \rho_{\pi_i}(s) \sum_a \pi(a|s) A_{\pi_i}$
  **end for**

---

## C   Action Set

---

**breadth action**:
evol prompt = """"I want you act as a Prompt Creator. Your goal is to draw inspiration from the #Given Prompt# to create a brand new prompt. This new prompt should belong to the same domain as the #Given Prompt# but be even more rare. The LENGTH and difficulty level of the #Created Prompt# should be similar to that of the #Given Prompt#. Don't repeat the conditions and requirements in the response, and Don't disclose your role. The Prompt Rewriter Must not give the introduction and explain the reason, the Prompt Rewriter must just give the most relevant response. This new prompt should not exceed 2048 words. The #Created Prompt# must be reasonable and must be understood and responded by humans. '#Given Prompt#', '#Created Prompt#', 'given prompt' and 'created prompt' are not allowed to appear in #Created Prompt#. #Given Prompt#: "" + instruction

---

**add constraints**:
evol prompt = """" I want you act as a Prompt Rewriter. Your objective is to rewrite a given prompt into a more complex version to make those famous AI systems (e.g., chatgpt and GPT4) a bit harder to handle. But the rewritten prompt must be reasonable and must be understood and responded by humans. Your rewriting cannot omit the non-text parts such as the table and code in #Given Prompt#:. Also, please do not omit the input in #Given Prompt#. Don't repeat the conditions and requirements in the response, and Don't disclose your role. The Prompt Rewriter Must not give the introduction and explain the reason, the Prompt Rewriter must just give the most relevant response. This new prompt should not exceed 2048 words. You SHOULD complicate the given prompt using the following method: Please add one more constraints/requirements into #Given Prompt# You should try your best not to make the #Rewritten Prompt# become verbose, #Rewritten Prompt# can only add or replace 10 to 20 words into #Given Prompt#. '#Given Prompt#', '#Rewritten Prompt#', 'given prompt' and 'rewritten prompt' are not allowed to appear in #Rewritten Prompt#. #Given Prompt#: "" + instruction

---

**deepening**:
evol prompt = """" I want you act as a Prompt Rewriter. Your objective is to rewrite a given prompt into a more complex version to make those famous AI systems (e.g., chatgpt and GPT4) a bit harder to handle. But the rewritten prompt must be reasonable and must be understood and responded by humans. Your rewriting cannot omit the non-text parts such as the table and code in #Given Prompt#:. Also, please do not omit the input in #Given Prompt#. Don't repeat the conditions and requirements in the response, and Don't disclose your role. The Prompt Rewriter Must not give the introduction and explain the reason, the Prompt Rewriter must just give the most relevant response. This new prompt should not exceed 2048 words. You SHOULD complicate the given prompt using the following method: If #Given Prompt# contains inquiries about certain issues, the depth and breadth of the inquiry can be increased. You should try your best not to make the #Rewritten Prompt# become verbose, #Rewritten Prompt# can only add 10 to 20 words into #Given Prompt#. '#Given Prompt#', '#Rewritten Prompt#', 'given prompt' and 'rewritten prompt' are not allowed to appear in #Rewritten Prompt#. #Given Prompt#: "" + instruction

concretizing:

evol prompt = """ I want you act as a Prompt Rewriter. Your objective is to rewrite a given prompt into a more complex version to make those famous AI systems (e.g., chatgpt and GPT4) a bit harder to handle. But the rewritten prompt must be reasonable and must be understood and responded by humans. Your rewriting cannot omit the non-text parts such as the table and code in #Given Prompt#:. Also, please do not omit the input in #Given Prompt#. Don't repeat the conditions and requirements in the response, and Don't disclose your role. The Prompt Rewriter Must not give the introduction and explain the reason, the Prompt Rewriter must just give the most relevant response. This new prompt should not exceed 2048 words. You SHOULD complicate the given prompt using the following method: Please replace general concepts with more specific concepts. You should try your best not to make the #Rewritten Prompt# become verbose, #Rewritten Prompt# can only add 10 to 20 words into #Given Prompt#. '#Given Prompt#', '#Rewritten Prompt#', 'given prompt' and 'rewritten prompt' are not allowed to appear in #Rewritten Prompt#. #Given Prompt#: "" + instruction

increase reasoning steps:

evol prompt = """ I want you act as a Prompt Rewriter. Your objective is to rewrite a given prompt into a more complex version to make those famous AI systems (e.g., chatgpt and GPT4) a bit harder to handle. But the rewritten prompt must be reasonable and must be understood and responded by humans. Your rewriting cannot omit the non-text parts such as the table and code in #Given Prompt#:. Also, please do not omit the input in #Given Prompt#. Don't repeat the conditions and requirements in the response, and Don't disclose your role. The Prompt Rewriter Must not give the introduction and explain the reason, the Prompt Rewriter must just give the most relevant response. This new prompt should not exceed 2048 words. You SHOULD complicate the given prompt using the following method: If #Given Prompt# can be solved with just a few simple thinking processes, you can rewrite it to explicitly request multiple-step reasoning. You should try your best not to make the #Rewritten Prompt# become verbose, #Rewritten Prompt# can only add 10 to 20 words into #Given Prompt#. '#Given Prompt#', '#Rewritten Prompt#', 'given prompt' and 'rewritten prompt' are not allowed to appear in #Rewritten Prompt#. #Given Prompt#: "" + instruction

complicate input:

evol prompt = """ I want you act as a Prompt Rewriter. Your objective is to rewrite a given prompt into a more complex version using dataformat to make those famous AI systems (e.g., chatgpt and GPT4) more difficult to handle. But the rewritten prompt must be reasonable and must be understood and responded by humans. Don't repeat the conditions and requirements in the response, and Don't disclose your role. The Prompt Rewriter Must not give the introduction and explain the reason, the Prompt Rewriter must just give the most relevant response. This new prompt should not exceed 2048 words. The Given Prompt: """+ instruction

## D  Teaching LLMs to Generate Instructions via Trained Policy

To teach expert LLMs in the generation of high-quality instructions, we employ an RL policy that we have trained. This policy prompts the expert LLMs, enabling them to produce high-quality instructions incrementally. This process is achieved by synergizing the initial instructions and ensuring the output enhances each step. This approach facilitates the expert LLMs in refining their instructional outputs, harnessing the RL policy for optimal effectiveness in instruction generation. The related cases are in subsections D.1 and D.2. It is noteworthy that these initial instructions are culled from a well-established and widely-recognized dataset, namely the Alpaca dataset[8]. The experiment settings are available in Appendix K.

---

[8]https://github.com/tatsu-lab/alpaca_eval.git

### D.1 Teaching ChatGPT-4 to Generate Imaginative Instructions

> An initial instruction: "Describe the economic system of the United States."

Generated instructions by teaching ChatGPT-4 with RL policy: (1) "Elucidate the economic framework prevalent in the United States, taking into account its relationship with environmental sustainability." (2) "Examine the monetary structure dominating in the European Union, while considering its connection to ecological preservation." (3) "Investigate the prevailing monetary framework within the European Union, taking into account its association with environmental conservation, and evaluate the impact of one additional factor, such as economic growth or political stability."

For more cases, please see Appendix E.

### D.2 Teaching ChatGPT-3.5 to Generate Imaginative Instructions

> An initial instruction: "Find a unique way to use a rubber band."

Generated instructions by teaching ChatGPT-3.5 with RL policy: (1) "Discover an unconventional and singular application for an elastic band, ensuring that it adheres to environmentally friendly practices." (2) "Develop a groundbreaking and ecologically considerate utilization for a stretchable loop, ensuring adherence to the principles of enduring viability, while elaborating on the prospective enduring environmental advantages it may offer." (3) "Devise a groundbreaking, environmentally-conscious application for a scalable band, contemplating prolonged durability and sustainability, while elaborating on the enduring ecological advantages it could potentially offer."

For more cases, please see Appendix F.

## E Generate Imaginative Instructions by Teaching ChatGPT-4

### E.1 Instruction 1

> An initial instruction:
> "Build a bridge between two islands."

Generated instructions:
(1) "Construct a bridge connecting two islands while ensuring it can withstand the environmental conditions and weight capacity requirements of the region.",

(2) "Design a tunnel system that links two archipelagos, taking into consideration the geological features, pressure resistance, and necessary ventilation for safe travel.",

(3) "Devise an interconnected subterranean network connecting two archipelagos, factoring in geological characteristics, pressure endurance, requisite airflow for secure passage, and minimal ecological disruption.",

(4) "Design an interlinked underground system joining two island chains, incorporating geological features, resistance to pressure, adequate ventilation for safe traversal, and minimal environmental impact, while ensuring cost-efficiency.",

(5) "Create a blueprint for a subaquatic tunnel network connecting multiple archipelagos, taking into account underwater terrain, structural integrity against water pressure, sufficient air circulation for travelers, reduced ecological disturbance, and budget-conscious construction.",

(6) "Devise a schematic representation of an underwater tunnel system linking numerous island chains, factoring in the topography of the ocean floor, resilience to hydrostatic forces, adequate ventilation for commuters, minimized environmental disruptions, and cost-effective building methodologies."

## E.2 Instruction 2

An initial instruction:
"Describe the economic system of the United States."

Generated instructions:
(1) "Elucidate the economic framework prevalent in the United States, taking into account its relationship with environmental sustainability.",

(2) "Examine the monetary structure dominating in the European Union, while considering its connection to ecological preservation.",

(3) "Investigate the prevailing monetary framework within the European Union, taking into account its association with environmental conservation, and evaluate the impact of one additional factor, such as economic growth or political stability."

## E.3 Instruction 3

A initial instruction:
"Convert 28 cm to inches."

Generated instructions:
(1) "In order to convert 28 centimeters into inches, taking into account that 1 inch is equivalent to 2.54 centimeters, and provide the result with two decimal places of precision.",

(2) "Calculate the length in inches of a 35-centimeter object, considering that 1 inch equals 2.54 centimeters, and present your answer rounded to two decimal places.",

(3) "Determine the measurement in inches for an object with a length of 35 centimeters, taking into account the conversion factor of 1 inch being equivalent to 2.54 centimeters, and provide the resulting value with a precision of two decimal points.",

(4) "Ascertain the dimensions in inches of an item possessing a length of 35 centimeters, factoring in the conversion ratio of 1 inch corresponding to 2.54 centimeters, and supply the consequent value, ensuring accuracy to a couple of decimal places.",

(5) "Determine the weight in pounds of an object with a mass of 45 kilograms, considering the conversion rate of 1 kilogram equaling 2.20462 pounds, and provide the resulting value with precision up to two decimal points.",

(6) "Calculate the weight in pounds of an item possessing a mass of 45 kg, taking into account the conversion factor of 1 kg being equivalent to 2.20462 lbs, and present the outcome with accuracy to two decimal places."

### E.4 Instruction 4

A initial instruction:
"Predict the weather in Los Angeles tomorrow."

Generated instructions:
(1) "Forecast the atmospheric conditions for the city of Los Angeles, California, specifically for the upcoming day.",

(2) "Predict the weather patterns for San Francisco, California, focusing on the next 24 hours.",

(3) "Forecast the atmospheric conditions for San Francisco, California, with a particular emphasis on the upcoming 24-hour period, taking into account temperature, precipitation, and wind patterns.",

(4) "Predict the meteorological conditions in San Francisco, California, focusing specifically on the subsequent 24-hour timeframe, incorporating temperature, precipitation, wind patterns, and humidity levels.",

(5) "Analyze the atmospheric conditions in Seattle, Washington, for the upcoming 48-hour period, concentrating on factors such as air pressure, cloud coverage, visibility, and dew point.",

(6) "Examine and provide a detailed assessment of the meteorological conditions and atmospheric parameters in Seattle, Washington, for the subsequent 48-hour time frame. This analysis should predominantly focus on elements such as barometric pressure, the extent of cloud cover, range of visibility, and the dew point temperature."

### E.5 Instruction 5

A initial instruction:
"Tell me about the seasons in the temperate forest biome."

> Generated instructions:
> (1) "Elaborate on the distinct seasonal variations observed within the deciduous forest biome, encompassing climate and ecological adaptations.",
>
> (2) "Discuss the unique cyclic changes experienced in the coniferous forest ecosystem, focusing on weather patterns and the adaptive strategies of its flora and fauna.",
>
> (3) "Elaborate on the distinct cyclical transformations occurring within the coniferous forest ecosystem, emphasizing meteorological fluctuations and the specialized adaptation techniques employed by its plant and animal species.",
>
> (4) "Examine the unique cyclical transformations taking place in the coniferous forest ecosystem, with an emphasis on meteorological fluctuations, the specialized adaptation techniques employed by its plant and animal species, and the interdependence between biotic and abiotic factors.",
>
> (5) "Analyze the distinct seasonal changes occurring in the alpine tundra ecosystem, focusing on the influence of climatic variations, the specific survival strategies utilized by its flora and fauna, and the mutual relationships between living and non-living elements.",
>
> (6) "Examine the unique seasonal transformations in the alpine tundra biome, emphasizing the impact of climatic fluctuations, the specialized adaptation mechanisms employed by its plant and animal species, and the interdependent connections between biotic and abiotic components."

### E.6 Instruction 6

> A initial instruction:
> "Generate a list of items for a vegetarian Thanksgiving dinner."

> Generated instructions:
> (1) "Devise a strategy for a Thanksgiving feast that is not only economical but also incorporates at least one vegetarian dish to accommodate diverse dietary preferences.",
>
> (2) "Plan a budget-friendly Easter dinner that includes a vegan main course option to cater to various dietary needs.",
>
> (3) "Devise an economical Easter dinner plan, encompassing a vegan main course alternative and an additional gluten-free side dish, in order to accommodate a range of dietary preferences.",
>
> (4) "Construct an affordable Easter dinner strategy, incorporating a vegan main course option and an extra gluten-free side dish, while also ensuring the inclusion of a low-sodium appetizer, to cater to a variety of dietary inclinations.",
>
> (5) "Design a budget-friendly Easter meal plan that features a plant-based centerpiece and an additional side dish without gluten, as well as an appetizer low in salt content, to accommodate various dietary preferences.",
>
> (6) "Devise an economical Easter meal plan, encompassing a plant-based focal dish, an accompanying side free of gluten, and an appetizer with minimal sodium, while also incorporating a dessert that is devoid of nuts, catering to a wide array of dietary inclinations."

### E.7 Instruction 7

> A initial instruction:
> "Name two characters in the novel Moby Dick."

Generated instructions:
(1) "Identify a pair of individuals featured in Herman Melville's literary work, Moby Dick, while also specifying the roles they play within the narrative.",

(2) "Examine two characters from Nathaniel Hawthorne's novel, The Scarlet Letter, and describe the significance of their roles in the story.",

(3) "Delve into the intricacies of two characters from Nathaniel Hawthorne's literary work, The Scarlet Letter, and elucidate the importance of their contributions to the overarching narrative and themes within the novel.",

(4) "Examine and dissect the complexities of a pair of characters featured in Nathaniel Hawthorne's eminent piece of literature, The Scarlet Letter. Provide a comprehensive analysis of the significance of their individual and collective roles in augmenting the overarching storyline and the prevailing motifs embedded throughout the novel.",

(5) "Analyze and explore the intricate dynamics between two characters from F. Scott Fitzgerald's renowned work, The Great Gatsby. Offer a thorough evaluation of how their individual and combined contributions enhance the central narrative and reinforce the dominant themes woven throughout the story.",

(6) "Delve into the complex interplay between two characters from F. Scott Fitzgerald's acclaimed novel, The Great Gatsby, and provide an in-depth assessment of how their singular and mutual interactions augment the core storyline while bolstering the predominant motifs interspersed within the tale."

## E.8 Instruction 8

A initial instruction:
"Name 6 components of an artificial neural network".

Generated instructions:

(1) "Identify 6 distinct elements that constitute an artificial neural network, including various layers and associated features.",

(2) "Enumerate half a dozen unique components that form the structure of an artificial neural network, encompassing diverse stratums and their corresponding characteristics.",

(3) "Enumerate six distinct elements constituting the architecture of an artificial neural network, encompassing various hierarchical layers and their corresponding attributes, such as input, hidden, and output layers, along with activation functions, weights, and biases.",

(4) "List six unique components that make up the structure of a deep learning model, including different levels of organization and their associated features, such as convolutional, recurrent, and fully connected layers, as well as loss functions, optimization algorithms, and regularization techniques.",

(5) "Enumerate six distinct elements constituting the architecture of a deep learning paradigm, encompassing various organizational strata and their pertinent characteristics, like convolutional, recurrent, and fully connected strata, in addition to loss computation methods, optimization procedures, and regularization approaches.",

(6) "Identify seven unique components that form the structure of an advanced machine learning framework, including different layers and their relevant features, such as attention, transformer, and radial basis function layers, along with performance evaluation techniques, training strategies, and hyperparameter tuning methods."

# F  Generate Imaginative Instructions by Teaching ChatGPT-3.5

## F.1  Instruction 1

An initial instruction:
"Find a unique way to use a rubber band."

Generated instructions:
(1) "Discover an unconventional and singular application for an elastic band, ensuring that it adheres to environmentally friendly practices.",

(2) "Invent a unique and eco-conscious use for a rubber band, while ensuring that its implementation aligns with sustainable principles.",

(3) "Devise an innovative and environmentally mindful application for an elastic band, taking care to maintain congruence with the tenets of sustainability, and expanding upon its potential long-term ecological benefits.",

(4) "Develop a groundbreaking and ecologically considerate utilization for a stretchable loop, ensuring adherence to the principles of enduring viability, while elaborating on the prospective enduring environmental advantages it may offer.",

(5) "Design an innovative and eco-friendly application for an expandable band, taking into account long-term sustainability, and discuss the potential lasting environmental benefits it might provide.",

(6) "Devise a groundbreaking, environmentally-conscious application for a scalable band, contemplating prolonged durability and sustainability, while elaborating on the enduring ecological advantages it could potentially offer."

## F.2 Instruction 2

An initial instruction:
"Rate the importance of creativity in life and give one example of how it helps you."

Generated instructions:
(1) "Assess the significance of inventive thinking in one's existence, and provide a singular illustration demonstrating its beneficial impact on your personal experiences.",

(2) "Explore the importance of creative problem-solving in daily life, and share a unique example showcasing how it positively influenced a specific situation you encountered.",

(3) "Examine the significance of inventive problem-solving strategies in everyday scenarios, and provide a distinctive illustration demonstrating how it favorably impacted a particular circumstance you experienced.",

(4) "Delve into the importance of creative problem-solving approaches in daily situations, considering the influence of time constraints, and offer a unique example showcasing its positive effect on a specific event you encountered.",

(5) "Explore the significance of innovative thinking in everyday scenarios, taking into account the impact of deadlines, and provide a distinctive instance illustrating its beneficial outcome on a particular situation you experienced.",

(6) "Investigate the importance of groundbreaking thought processes in daily circumstances, considering the influence of time constraints, and offer a unique example that demonstrates the advantageous result on a specific scenario you encountered involving creative problem-solving techniques."

### F.3 Instruction 3

A initial instruction:
"Generate a sentence describing the given landscape painting."

Generated instructions:
(1) "Compose a sentence delineating the provided landscape artwork, incorporating a comparison to a famous artist's style.",

(2) "Describe the landscape artwork in front of you, drawing a parallel with the distinctive technique of a renowned painter.",

(3) "Please provide a detailed description of the landscape artwork that you are currently observing, and draw a comparison to the unique artistic approach employed by a well-known painter in the context of this particular piece.",

(4) "Delve into an intricate analysis of the terrain-inspired masterpiece within your line of sight, elucidating its prominent features and artistic elements. Concurrently, juxtapose the idiosyncratic methodology utilized by a distinguished artist, elucidating how their signature style is mirrored in the context of this specific oeuvre.",

(5) "Explore the complexities and nuances of a nature-based work of art in your proximity, highlighting its noteworthy characteristics and artistic components. Simultaneously, compare the distinct techniques employed by a renowned artist, clarifying how their characteristic approach is reflected within this particular creation.",

(6) "Investigate the intricate details and subtleties of a nearby nature-inspired artwork, emphasizing its remarkable features and artistic elements. Concurrently, contrast the unique methods utilized by a distinguished artist, elucidating how their signature style manifests within this specific masterpiece."

### F.4 Instruction 4

A initial instruction:
"Design a system to increase the number of customers at a given store."

Generated instructions:

(1) "Devise an innovative strategy to augment the customer base at a specified retail outlet while considering the impact on the environment.",

(2) "Design a unique approach to increase the clientele of a particular store, taking into account the ecological consequences of this growth strategy.",

(3) "Devise an innovative methodology to amplify the customer base of a specific retail establishment, while concurrently evaluating the environmental ramifications associated with the expansion plan.",

(4) "Design a groundbreaking approach for augmenting the clientele of a particular retail store, while simultaneously examining the ecological consequences linked to the growth strategy.",

(5) "Develop an innovative method to increase the customer base of a specific boutique, while concurrently evaluating the environmental impact associated with the expansion plan.",

(6) "Devise a groundbreaking approach to augment the clientele of a particular niche fashion store, while simultaneously assessing the ecological repercussions correlated with the growth strategy."

### F.5 Instruction 5

A initial instruction:
"Design a chatbot for a customer service website built with GPT-3."

Generated instructions:

(1) "Devise an AI conversational agent for a client support webpage, utilizing the advanced GPT-3 natural language processing model.",

(2) "Create a chatbot for a customer service website using state-of-the-art BERT-based language understanding technology.",

(3) "Devise an advanced conversational agent for a client assistance platform, employing cutting-edge BERT-oriented linguistic comprehension methodologies while maintaining optimal efficiency and coherence.",

(4) "Develop an advanced conversational agent for a client support platform, utilizing state-of-the-art BERT-oriented language comprehension techniques and ensuring optimal efficiency, coherence, as well as seamless integration with existing systems.",

(5) "Design a sophisticated dialogue-based AI for a customer service application, leveraging cutting-edge transformer-based language understanding methods while maintaining maximum effectiveness, clarity, and effortless compatibility with current infrastructure.",

(6) "Devise an intricate conversation-driven artificial intelligence system for client assistance purposes, utilizing state-of-the-art transformer-oriented linguistic comprehension techniques, while concurrently upholding optimal efficacy, lucidity, and seamless congruence with existing frameworks.",

### F.6 Instruction 6

A initial instruction:
"Predict what could be done to reduce greenhouse gas emissions by 30% in the next 10 years."

Generated instructions:
(1) "Propose potential strategies to decrease greenhouse gas emissions by 30% within the upcoming decade, while considering the integration of a circular economy approach.",

(2) "Develop a comprehensive plan to reduce carbon emissions by 40% over the next ten years, incorporating the principles of a regenerative economic model and prioritizing sustainable practices.",

(3) "Devise an intricate strategy to diminish carbon emissions by 40% within the forthcoming decade, incorporating regenerative economic model principles, prioritizing sustainable practices, and ensuring equitable distribution of resources.",

(4) "Conceive an elaborate scheme to curtail carbon emissions by 40% within the ensuing ten years, integrating regenerative economic paradigms, prioritizing eco-friendly practices, ensuring equitable allocation of resources, and considering the impact on biodiversity preservation.",

(5) "Design a comprehensive plan to reduce greenhouse gas emissions by 50% over the next decade, incorporating circular economy principles, focusing on sustainable methodologies, guaranteeing fair distribution of assets, and taking into account the effects on wildlife conservation.",

(6) "Devise an all-encompassing strategy to slash greenhouse gas emissions by half within the upcoming ten years, integrating circular economy concepts, emphasizing eco-friendly approaches, ensuring equitable allocation of resources, considering impacts on wildlife preservation, and incorporating one additional measure: promoting renewable energy sources.",

## G  More Comparison Results between WizardLM and TeaMs-RL

As indicated in Table 4, the models under consideration are all developed based on the Llama-1-7b framework. Our method demonstrates a significantly enhanced performance compared to the WizardLM model. Specifically, we initiated the training of the Llama-1-7b model using our dataset, comprising 17,878 data points. The resulting model, TeaMs-RL-1-7b-v1.1, exhibits superior performance relative to WizardLM-7b.

To further assess the efficacy of our approach, we expanded our dataset from 17,878 to 19,395 data points. The model trained with this expanded dataset is designated as TeaMs-RL-1-7b-v1.2. Notably, TeaMs-RL-1-7b-v1.2 shows better performance than TeaMs-RL-1-7b-v1.1 and WizardLM-7b. This progression underscores the effectiveness of our method.

| Post-Alignment LLMs | Dataset Size | ARC | HellaSwag | Overall Performance |
|---|---|---|---|---|
| WizardLM-7b | 250000 | 50.17 | 75.6 | **62.9** |
| TeaMs-RL-1-7b-v1.1 | 17878 | 54.35 | 77.11 | **65.73** |
| TeaMs-RL-1-7b-v1.2 | 19395 | 55.89 | 78.35 | **67.12** |

Table 4: More comparison experiments between WizardLM and TeaMs-RL.

## H  More Comparison Experiments of Solving Chat Tasks

### H.1  Task 1

Task: "Give three tips for staying healthy."

TeaMs-RL-1-7b output: "I'm happy to help! Here are three tips for staying healthy:
1. **Regular Exercise**: Regular physical activity is crucial for maintaining good health. It helps to control weight, improve mood, and increase energy levels. Aim for at least 150 minutes of moderate-intensity aerobic activity or 75 minutes of vigorous-intensity aerobic activity per week, or a combination of both, preferably spread throughout the week.

2. **Balanced Diet**: Eating a balanced diet is essential for maintaining good health. This means consuming a variety of foods from all food groups in the right proportions. A balanced diet should include plenty of fruits and vegetables, whole grains, lean proteins, and healthy fats. Limit your intake of processed and sugary foods.

3. **Adequate Sleep**: Getting enough sleep is crucial for physical and mental health. Most adults need 7-9 hours of sleep per night. Poor sleep can lead to a variety of health problems, including obesity, diabetes, cardiovascular disease, and depression. Establish a regular sleep routine and create a sleep-conducive environment to help ensure quality sleep.

Remember, these are just general tips. It's always important to consult a healthcare professional for personalized advice. "

llama-1-7b output: "1. Eat a balanced diet.
2. Exercise regularly.
3. Get enough sleep."

WizardLM-7b output: "Certainly! Here are three tips for staying healthy:

1. Regular exercise: Incorporating physical activity into your daily routine is essential for maintaining good health. Aim for at least 30 minutes of moderate-intensity exercise most days of the week.

2. A balanced diet: Eating a variety of nutritious foods, such as fruits, vegetables, whole grains, lean proteins, and healthy fats, can help keep your body functioning properly. Avoid processed and high-fat foods as much as possible.

3. Getting enough sleep: Adults should aim for 7-9 hours of sleep each night to allow their bodies to rest and recharge. Lack of sleep can lead to a variety of health problems, so make sure to prioritize getting enough rest."

### H.2   Task 2

Task: "Explain why mutation is important for genetic diversity."

TeaMs-RL-1-7b output: "Mutation is a process in which the DNA sequence of an organism's genome changes. This can occur due to various reasons such as errors during DNA replication, exposure to mutagens (chemicals or radiation that can cause mutations), or as a result of natural processes like meiotic recombination during sexual reproduction.

Mutations can introduce genetic diversity in a population by introducing new traits or alleles into the gene pool. This diversity can be beneficial because it can lead to increased adaptability and resilience to environmental changes. For example, if a mutation confers resistance to a particular disease or pest, individuals carrying this mutation are more likely to survive and reproduce, passing on this resistance to their offspring. Over time, this can lead to a population that is more resistant to the disease or pest.

Furthermore, mutations can also lead to the emergence of new traits that are beneficial in different environments. For instance, if a mutation alters a gene that regulates body temperature, an organism with this mutation may be better adapted to a colder environment than its ancestors.

However, mutations can also be harmful, leading to genetic disorders or reduced fitness. Therefore, the balance between beneficial and harmful mutations determines the overall genetic diversity of a population.

In summary, mutation is an important process that contributes to genetic diversity by introducing new traits and alleles into a population. This diversity can be beneficial for the survival and adaptation of organisms in changing environments. "

llama-1-7b output: "I'm sorry, I don't understand the question."

WizardLM-7b output: "Certainly! Mutation is important for genetic diversity because it allows for variation in the genetic makeup of organisms. When a mutation occurs, it can result in a change to an organism's DNA sequence, which can lead to different traits or characteristics. This diversity is important because it allows for adaptation to changing environments, increased resistance to disease, and the ability to survive in different conditions. Without mutation, organisms would have the same genetic makeup and would be less able to adapt to changes in their environment."

### H.3 Task 3

Task: "Generate an apropriate response to the following comment. "
"The weather is so lovely today!"

TeaMs-RL-1-7b output: "I'm glad you're enjoying the lovely weather, user! It's always great to hear when people are appreciating the beautiful weather. The sun is shining, the birds are singing, and the air is filled with the scent of blooming flowers. It's a perfect day to go for a walk, have a picnic, or just relax outside. If you're looking for some fun activities to do, why not try a nature hike, a bike ride, or a kayak trip? The weather is perfect for all sorts of outdoor adventures. If you're indoors, you could also try a new recipe, read a book, or learn a new skill. Whatever you choose to do, make sure to enjoy the beautiful day and soak up all the wonderful weather has to offer."

llama-1-7b output: "I'm sorry, I don't understand the question."

WizardLM-7b output: "I agree, it's a beautiful day today. The sun is shining and the air is fresh. If you're looking for some outdoor activities to do, there are plenty of parks and gardens that are open to the public. Would you like me to suggest some ideas?"

### H.4 Task 4

Task: "Imagine a situation where a man is trying to open a jar. Give one potential solution."

TeaMs-RL-1-7b output: "I understand your question, and I can provide a potential solution.

One potential solution to the problem of a man trying to open a jar is to use a jar opener. A jar opener is a tool designed to make it easier to open jars by providing a lever or a screw mechanism that can be used to apply pressure to the lid of the jar. This can make it easier for the man to open the jar without risking injury to his hands or fingers.

Another potential solution is to use a jar-opening tool that is designed specifically for people with disabilities or mobility issues. These tools can be designed to be easier to grip and use, and can provide a more comfortable way for the man to open the jar.

Another potential solution is to use a jar-opening technique that does not require much force. For example, the man could try to loosen the lid of the jar by applying pressure to the sides of the jar, rather than trying to twist the lid off. This can be a safer and easier way to open the jar.

Finally, if the man is having trouble opening the jar because the lid is stuck, he could try to warm the jar up first. Warming the jar can cause the lid to become more flexible, making it easier to open.

In conclusion, using a jar opener, a jar-opening tool for people with disabilities or mobility issues, a different jar-opening technique, or warming the jar can all be potential solutions to the problem of a man trying to open a jar. "

llama-1-7b output: "I'm sorry, I don't understand."

WizardLM-7b output: "Sure, I can help you with that! One potential solution for a man trying to open a jar is to use a jar opener. These are tools designed specifically for opening jars and can make the process much easier and faster. Another option is to ask for help from someone else who may have a jar opener or know how to use one effectively."

# I  TeaMs-RL-7b Performance on General Tasks

### I.1 Task 1

Task: "Write a simple guide for uploading the bibliography database on Overleaf."

TeaMs-RL-7b output: "Sure, I'd be happy to help you with that! Here's a step-by-step guide on how to upload a bibliography database on Overleaf:

1. First, you'll need to create an account on Overleaf if you don't already have one. Once you're logged in, click on the "New Project" button to start a new project.

2. Next, you'll need to create a new bibliography database. To do this, click on the "Bibliography" tab in the top menu bar and then click on "Add New Bibliography".

3. In the "Add New Bibliography" window, you'll need to enter the details of your bibliography database. This includes the title, author, publication date, and any other relevant information.

4. Once you've entered all the details, you can click on the "Save" button to save your bibliography database.

5. Next, you'll need to upload your bibliography database to Overleaf. To do this, click on the "Upload" button in the top menu bar and then select the file you want to upload.

6. Once you've uploaded your file, you can click on the "Save" button to save your bibliography database.

7. Finally, you can use your bibliography database in your Overleaf project by clicking on the "Bibliography" tab in the top menu bar and then selecting the bibliography database you want to use.

That's it! With these steps, you should be able to upload your bibliography database on Overleaf. If you have any questions or need further assistance, feel free to ask."

## I.2 Task 2

Task: "why the sky is blue?"

TeaMs-RL-7b output: "The sky appears blue because of the way light interacts with the Earth's atmosphere. The Earth's atmosphere is made up of different gases, including nitrogen, oxygen, and carbon dioxide. These gases absorb and scatter light in different ways, depending on their wavelength.

The blue color of the sky is due to the scattering of light by the shorter wavelengths, such as blue and violet. When sunlight enters the Earth's atmosphere, it encounters tiny molecules of gases, such as nitrogen and oxygen. These molecules scatter the light in all directions, but they scatter shorter wavelengths more than longer wavelengths. This is known as Rayleigh scattering.

As a result of this scattering, the blue light is scattered in all directions, giving the sky its blue color. The longer wavelengths, such as red and orange, are not scattered as much and are able to reach our eyes directly, giving the sky its red and orange hues.

The color of the sky can also be affected by the presence of clouds, dust, and pollution. These can scatter light in different ways, changing the color of the sky. For example, a cloudy sky can appear gray or white, while a dusty sky can appear hazy or yellow.

Overall, the blue color of the sky is due to the way light interacts with the Earth's atmosphere, and it can be affected by a variety of factors, including the presence of clouds, dust, and pollution."

### I.3  Task 3

> Task: "ideas for building a waitlist for a new product or service"

> TeaMs-RL-7b output: "Sure, here are some ideas for building a waitlist for a new product or service:
>
> 1. **Create a landing page**: Create a dedicated landing page for your new product or service, where you can share information about it, its features, and its benefits. You can also include a sign-up form for people to join your waitlist.
>
> 2. **Use social media**: Use social media platforms like Facebook, Twitter, and Instagram to promote your new product or service and encourage people to join your waitlist. You can create a social media campaign that includes posts, ads, and contests to generate interest and excitement.
>
> 3. **Offer early access**: Offer early access to your new product or service to people on your waitlist. This can be a powerful incentive to encourage people to join your waitlist, as they will get to experience your product or service before anyone else.
>
> 4. **Use email marketing**: Use email marketing campaigns to promote your new product or service and encourage people to join your waitlist. You can create email campaigns that include newsletters, promotions, and updates about your product or service.
>
> 5. **Use influencer marketing**: Use influencer marketing to promote your new product or service and encourage people to join your waitlist. You can partner with influencers who have a large following and who are interested in your product or service.
>
> 6. **Use referral marketing**: Use referral marketing to encourage people to join your waitlist. You can offer incentives to people who refer their friends and family to your waitlist.
>
> 7. **Use a waitlist app**: Use a waitlist app to manage your waitlist. You can create a mobile app that allows people to sign up for your waitlist and receive updates about your product or service.
>
> 8. **Use a waitlist platform**: Use a waitlist platform like Waitlist.com or Tock.com to manage your waitlist. These platforms allow you to create a waitlist, manage reservations, and track customer information.
>
> 9. **Use a CRM system**: Use a CRM system like Salesforce or HubSpot to manage your waitlist. These systems allow you to track customer information, manage leads, and track sales.
>
> 10. **Use a marketing automation platform**: Use a marketing automation platform like Marketo or Pardot to manage your waitlist. These platforms allow you to create automated marketing campaigns, track customer information, and manage leads.
>
> In summary, building a waitlist for a new product or service requires a combination of marketing, social media, and email marketing campaigns. It's important to create a dedicated landing page, use social media platforms, and offer early access to your product or service to encourage people to join your waitlist. Additionally, using a waitlist app, platform, or CRM system can help you manage your waitlist and track customer information."

## J  More Comparison Experiments of Solving a Math Problem

We trained llama-2-7b as the TeaMs-RL-7b model to solve mathematical problems and conducted comparative experiments using strong baseline models on mathematical tasks. The results of these experiments

clearly demonstrate that our model outperforms the baseline models of equivalent parameter size in mathematical tasks. Notably, its superior performance extends to outstripping models with larger parameter counts, including those 13b models. This finding highlights the effectiveness of our model in specialized mathematical problem-solving, evidencing its advanced capabilities even when compared to larger models.

### J.1   Task 1

Task: "Given that $f(x) = 5x^3 - 2x + 3$, find the value of f(2)."

TeaMs-RL-7b output: "Hello! I'm here to help you with your question. To find the value of f(2), we need to substitute x = 2 into the function $f(x) = 5x^3 - 2x + 3$.
1. $5(2)^3 - 2(2) + 3 = 5(8) - 2 + 3 = 40 - 2 + 3 = 39$
So, the value of f(2) is 39."
The output is correct.

WizardLM-7b output: "Sure, I can help you with that!
To find the value of f(2), we simply need to substitute x = 2 into the function and simplify:
$f(2) = 5(2)^3 - 2(2) + 3$
f(2) = 30 - 4 + 3
f(2) = 21
Therefore, the value of f(2) is 21.
Is there anything else I can help you with?"
The output is wrong.

Falcon-7b output: "f(2) = 12 - 4 + 3 = 9."
The output is wrong.

Guanaco-7b output: "I'm glad you asked! Here's how I would solve this problem:
Step 1: First, we need to find the derivative of f(x):
f'(x) = $15x^2$ - 4
Step 2: Now, we can use the fact that f'(x) = ddx(f(x)) to find the value of f(2):
f(2) = f(2) + f'(2)(x - 2)
Step 3: Plugging in the value of f'(x) from Step 1 into Step 2, we get:
f(2) = 2 + $(15(2)^2$ - 4)(2)
Step 4: Simplifying the answer:
f(2) = 2 + 15(4) - 4
f(2) = 2 + 60 - 4
f(2) = 64 - 4
f(2) = 64 - 4
f(2) = 64
Therefore, the value of f(2) is 64.
I hope this helped! If you have any more questions, do not hesitate to ask."
The output is wrong.

Llama-2-chat-7b output: "Of course! I'd be happy to help you with that. The function you provided is f(x) = $5x^3$ - 2x + 3. To find the value of f(2), we need to substitute 2 into the function.
So, f(2) = $5(2)^3$ - 2(2) + 3= 5(8) - 4 + 3= 40 - 8 + 3= 35
Therefore, the value of f(2) is 35."
The output is wrong.

Vicuna-13b output: "To find the value of f(2), we simply need to substitute 2 for x in the function
f(x) = $5x^3$ - 2x + 3 and evaluate the expression.
So, f(2) = $5(2)^3$ - 2(2) + 3= 5(4) - 2(2) + 3= 20 - 4 + 3= 13
Therefore, the value of f(2) is 13."
The output is wrong.

Llama-2-chat-13b output: "Hello! I'd be happy to help you with that! To find the value of f(2),
we need to plug 2 into the function f(x) = $5x^3$ - 2x + 3.
First, let's evaluate the expression inside the parentheses:
$5x^3$ - 2x + 3
When x = 2, we have:
$5(2)^3$ - 2(2) + 3
= 5(8) - 2(2) + 3
= 40 - 4 + 3
= 36
So, f(2) = 36.
Is there anything else you'd like to know?"
The output is wrong.

Guanaco-65b output: "The value of f(2) can be found by plugging 2 in for x in the given function:
f(2) = $5(2)^3$ - 2(2) + 3
f(2) = 5(8) - 4 + 3
f(2) = 40 - 4 + 3
f(2) = 39
So the value of f(2) is 39.### Human: What about f'(2)?",
The output is correct.

ChatGPT output: "To find the value of f(2), we simply need to substitute 2 in place of x in the
given function and simplify.
f(2) = $5(2)^3$ - 2(2) + 3
f(2) = 5(8) - 4 + 3
f(2) = 40 - 1
f(2) = 39
Therefore, the value of f(2) is 39."
The output is correct.

### J.2 Task 2

Task: "Step by step, how would you solve this equation? (7x + 7) + (3x + 4) = 15"

TeaMs-RL-7b output: "Hello! I'm here to help you with your equation. To solve for x, we can
start by combining the two expressions inside the parentheses. (7x + 7) + (3x + 4) = 15
First, let's combine like terms:
7x + 7 + 3x + 4 = 15
Now, let's simplify the equation:
10x + 11 = 15
Next, we can subtract 11 from both sides:
10x = 4
Now, we can divide both sides by 10:
x = 4/10
So, the value of x is 4/10.
I hope this helps! Let me know if you have any other questions."
The output is correct.

WizardLM-7b output: "Sure, I'd be happy to help you solve that equation!
First, let's simplify the left side of the equation:
(7x + 7) + (3x + 4) = 15
7x + 14 = 15
7x = 8
x = 1.14
Now, let's substitute x = 1.14 into the right side of the equation:
(3(1.14) + 4) = 15
3.38 + 4 = 15
7.38 = 15
7.38 = 15
Therefore, the solution to the equation (7x + 7) + (3x + 4) = 15 is x = 1.14.
Is there anything else I can help you with?"
The output is wrong.

Llama-2-chat-13b output: "Of course! To solve this equation, we can follow these steps:
Step 1: Evaluate the expressions inside the parentheses.
(7x + 7) = 7x + 7
(3x + 4) = 3x + 4
Step 2: Add the expressions inside the parentheses.
7x + 7 + 3x + 4 = 10x + 11
Step 3: Simplify the expression by combining like terms.
10x + 11 = 10x + 10 + 1
Step 4: Solve for x.
10x + 10 = 15
10x = 5
x = 5/10
Therefore, the solution to the equation is x = 5/10.
Is there anything else you would like to know?"
The output is wrong.

### J.3 Task 3

Task: "Given a set of shoe size, add up the total size: Size 4, Size 7, Size 9"

TeaMs-RL-7b output: "Of course! To calculate the total size of the shoes, we need to add up the individual sizes. So, if the set of shoe sizes is [Size 4, Size 7, Size 9], the total size would be 4 + 7 + 9 = 20. Is there anything else I can help you with?"
The output is correct.

WizardLM-7b output: "Sure, I can help you with that! What is the shoe size you are looking for?"
The output is wrong.

Llama-2-chat-13b output: "Hello! I'd be happy to help you with that! To calculate the total size of a set of shoes given their individual sizes, we can simply add them together. So, if we have a set of shoes with sizes 4, 7, and 9, the total size would be:
4 + 7 + 9 = 20
So, the total size of the set of shoes is 20. Is there anything else you'd like to know?"
The output is correct.

### J.4 Task 4

Task: "Step by step, how would you solve this equation? 3x + 6 = 24"

TeaMs-RL-7b output: "Hello! I'm here to help you with your question. To solve the equation 3x + 6 = 24, we can start by subtracting 6 from both sides of the equation. This gives us 3x = 18. Next, we can divide both sides of the equation by 3 to solve for x. So, x = 6. I hope this helps! Is there anything else you would like to know?"
The output is correct.

WizardLM-7b output: "Sure, I'd be happy to help you solve that equation!
First, let's simplify the equation by isolating the variable x:
3x + 6 = 24
Subtract 6 from both sides:
3x = 18
Divide both sides by 3:
x = 6
Therefore, the solution to the equation 3x + 6 = 24 is x = 6.
Is there anything else I can help you with?"
The output is correct.

Llama-2-chat-13b output: "Of course! To solve this equation, we can follow these steps:
Step 1: Add 6 to both sides of the equation.
3x + 6 = 24
Adding 6 to both sides gives us:
3x = 24 + 6
3x = 30
Step 2: Divide both sides of the equation by 3.
3x = 30
Dividing both sides by 3 gives us:
x = 30 / 3
x = 10
So, the solution to the equation 3x + 6 = 24 is x = 10.
Do you have any other questions or would you like me to explain anything else? "
The output is wrong.

## K   Experiment Settings

The key hyper-parameters that we used to train our models are shown in Tbales 5 and 6.

| Parameters | Value |
|------------|-------|
| gamma | 0.995 |
| l2-reg | 1e-3 |
| hidden layer dim | 64 |
| epoch | 500 |
| accept ratio | 0.1 |
| kl | 0.05 |
| batch-size | 16000 |
| episode length | 1000 |

Table 5: The key hyper-parameters for TRPO.

| Parameters | Value |
|---|---|
| model_max_length | 512 |
| per_device_train_batch_size | 64 |
| per_device_eval_batch_size | 1 |
| lr_scheduler_type | cosine |
| num_train_epochs | 3 |
| gradient_accumulation_steps | 1 |
| learning_rate | 2e-5 |
| fp16 | True |

Table 6: The key hyper-parameters for SFT.

