# OpenReview forum: "TeaMs-RL: Teaching LLMs to Generate Better Instruction Datasets via Reinforcement Learning"
_TMLR — Accepted by TMLR_

### Review · Reviewer_9P7n · 2024-08-26

**Summary Of Contributions:**

This paper introduces TeaMs-RL, a novel approach for generating instruction datasets for tuning Large Language Models (LLMs).
TeaMs-RL follows a reinforcement learning approach rather than relying on human feedback for generating.
Moreover, the access to expert LLMs (costly and energy-intensive) or reduced in comparison to the current practice of self-instruction.
For this, TeaMs-RL uses reinforcement learning to learn a policy which allows models, such as ChatGPT, to generate high quality instructions.
Compared to existing models, TeaMs-RL achieves better performance with a smaller fine-tuning dataset.

**Audience:**

Yes

**Broader Impact Concerns:**

No concerns

**Claims And Evidence:**

Yes

**Requested Changes:**

Please see “Strengths and Weaknesses” for more details.
1. Provide more details (what and why)
	- Tasks
	- Metrics
	- Models


2. Better positioning within RW, in particular other self-instruct approaches

3. Consider re-designing Section 4


Small remarks:
- Please clarify what it means to “teach” an expert LLM. Is the expert changed in the process?

- Some of the terms appear quite advanced, at least to me, and makes it difficult to understand their meaning:judicial, congruence, delineated

- Page 5 “..”

- Page 3 talks about “transferability”, while possible not tested for other models than Llama, so potentially consider limitations to efficiency when transferred to other models

**Strengths And Weaknesses:**

Strengths:

- Addresses an important problem of teaching LLMs how to generate instruction datasets for model training at a lower cost.
- Improvements in multiple dimensions, which includes performance on several tasks, queries to expert LLMs and thereby sustainability.
- Well written. At times it feels like simpler words could have been chosen to make the paper easier to read.
- Code is shared. This includes experimental scripts. It could be further improved with more instructions and potentially the inclusion of trained models.

Weaknesses:


- Overall, there is little explanation provided for several aspects. This includes model choices, benchmarking tasks, metrics and related works. For instance, the related work section is missing a comparison with other self-instruct approaches. The main focus was put on WizardLM, while other approaches have been omitted. Rather, they were grouped as “prevailing self-instruct methods” without further references or details. Privacy concerns are mentioned very briefly and it could be a consideration to either provide more details on why it is important or focus on other benchmarks altogether. Additionally, it could be beneficial to outline certain aspects in more detail, such as TRPO.

- The benchmark selection of Wizard7B could be better motivated. It is important to show why it has been chosen and where it is has been downloaded. The WizardLM publication presents several models and their performance on 9 tasks (including ARC and HellaSwag) all of which perform better than TeaMs-RL. Only one of the models is 7B and is based on Mistral, which could make a comparison more difficult. However, details on why Llama was chosen as a base and only a 7B WizardLM version used for benchmarking is needed. More importantly, while a link to the 13B model is provided, it is missing for the 7B version.

- Section 4 appears unstructured. There is information missing why certain tasks and metrics have been chosen for comparison, why they are presented in different subsection and are not presented in the same tables. A clearer outline on what is to be investigated, what the tasks are and the corresponding metrics would help to better follow it. For instance, 4.3 and 4.5 consider tasks which are also used in WizardLM (which uses a total of 9 tasks). Section 4.2, 4.6 and 4.7 seem qualitative, with single examples shown. Section 4.1. considers diversity, but the metric should be explained in more detail and motivated why it has been chosen over other ones.

---

> ### Author Response · Authors · 2024-10-21
>
> > **Q1:** Overall, there is little explanation provided for several aspects. This includes model choices, benchmarking tasks, metrics and related works. For instance, the related work section is missing a comparison with other self-instruct approaches. The main focus was put on WizardLM, while other approaches have been omitted. Rather, they were grouped as “prevailing self-instruct methods” without further references or details. Privacy concerns are mentioned very briefly and it could be a consideration to either provide more details on why it is important or focus on other benchmarks altogether. Additionally, it could be beneficial to outline certain aspects in more detail, such as TRPO.
> - **A1:**  We appreciate the reviewer's valuable comments. Regarding model choices, due to limited computational resources, we deploy our methods on 7B models. We select WizardLM as the baseline, which is a state-of-the-art method. For metrics, we utilize widely adopted benchmarks to evaluate the effectiveness of our approach.
> - In terms of related work, we will revise the section to include a broader range of studies. If the reviewer could provide specific references, we would greatly appreciate it, as it would help us present a more thorough discussion of related self-instruct approaches.
> - As for privacy concerns, we see this as a potential strength of our method, demonstrating its broader applicability. We plan to further extend our work to safety and privacy-focused applications to showcase its effectiveness in these domains.
> - Additionally, we have provided a more detailed explanation of TRPO in the appendix, and the related code can be accessed via the link: [https://anonymous.4open.science/r/TeaMs-RL-D3FE/Teams_RL_GPT/teams_rl/algorithms/trpo.py](https://anonymous.4open.science/r/TeaMs-RL-D3FE/Teams_RL_GPT/teams_rl/algorithms/trpo.py).
>
>
>
> > **Q2:** The benchmark selection of Wizard7B could be better motivated. It is important to show why it has been chosen and where it is has been downloaded. The WizardLM publication presents several models and their performance on 9 tasks (including ARC and HellaSwag) all of which perform better than TeaMs-RL. Only one of the models is 7B and is based on Mistral, which could make a comparison more difficult. However, details on why Llama was chosen as a base and only a 7B WizardLM version used for benchmarking is needed. More importantly, while a link to the 13B model is provided, it is missing for the 7B version.
> - **A2:** Thank you for your insightful comments. We have addressed your concerns as follows:
> - **Model, Benchmark Selection and Motivation**: Due to limited computational resources, we chose to benchmark against the 7B version of WizardLM to ensure a fair comparison with our own model, which is also based on a 7B parameter size.  Note, in all our experiments, due to computational resource constraints, all models are configured with a float16 format using the same settings, rather than 32-bit float, as this can save memory. The experimental results may differ from those reported in the original baseline paper.
>
> - **Choice of LLaMA as Base Model**: We use LLaMA as our base model due to its proven effectiveness and widespread adoption in the research community. LLaMA models offer a good balance between performance and resource requirements, making them suitable for our computational constraints. Additionally, using LLaMA facilitates reproducibility and comparison with other studies that have employed the same base.
>
> - **Download Source of WizardLM-7B**: We have updated the paper to include the download source: [WizardLM-7B on Hugging Face](https://huggingface.co/TheBloke/wizardLM-7B-HF).

---

> ### Author Response · Authors · 2024-10-21
>
> > **Q3:**  Section 4 appears unstructured. Certain tasks and metrics have been chosen for comparison, why they are presented in different subsection and are not presented in the same tables. A clearer outline on what is to be investigated, what the tasks are and the corresponding metrics would help to better follow it. For instance, 4.3 and 4.5 consider tasks which are also used in WizardLM (which uses a total of 9 tasks). Section 4.2, 4.6 and 4.7 seem qualitative, with single examples shown. Section 4.1. considers diversity, but the metric should be explained in more detail and motivated why it has been chosen over other ones.
> - **A3:** Thank you for your useful feedback. Below, we address each of your concerns:
>
> - **Overall Structure of Section 4**: We understand that the original organization may have appeared unstructured. In the revised manuscript, we have restructured Section 4 to provide a clearer and more logical flow. We now begin with an overview of our experimental setup. This reorganization is intended to guide the reader through our investigation systematically. **Subsection 4.1 (Diversity Analysis)**: This section analyzes the diversity of policies generated by our method. **Subsection 4.2 (Instruction Generation)**: Here, we demonstrate how our method generates diverse instructions (We have moved this section to Appendix). **Subsection 4.3 (Performance on Standard Benchmarks)**: We evaluate our trained models against standard benchmarks, comparing them with other state-of-the-art foundation models, including those used in WizardLM. **Subsection 4.4 (Privacy Preservation)**: This section highlights the privacy-preserving features of our method. **Subsections 4.5 to 4.7 (Additional Evaluations)**: We further assess our method using various tasks, including general knowledge and mathematical reasoning tasks.
> - **Qualitative vs. Quantitative Analysis**: We presented quantitative experiments in Subsections 4.1, 4.3, 4.4, and 4.5. To further demonstrate the effectiveness of our method, we included qualitative experiments in Subsections 4.2, 4.6, and 4.7. Together, these results indicate that our conclusions are well-supported by empirical evidence.
>
> - **Explanation of the Diversity Metric**: Action diversity refers to the breadth of information coverage; higher diversity indicates more comprehensive information. Therefore, we use action diversity as a metric to assess policy quality. The experimental results further validate the effectiveness of this metric.
>
>
> > **Q4:** Better positioning within RW, in particular other self-instruct approaches
> - **A4:**  We apologize for the confusion, as we are unsure of the specific meaning of "RW" in this context. Could the reviewer kindly provide further clarification or details?
>
> > **Q5:** Small remarks: (1) Please clarify what it means to “teach” an expert LLM. Is the expert changed in the process? (2) Some of the terms appear quite advanced, at least to me, and makes it difficult to understand their meaning:judicial, congruence, delineated. (3) Page 5 “..” (4) Page 3 talks about “transferability”, while possible not tested for other models than Llama, so potentially consider limitations to efficiency when transferred to other models
> - **A5:** Thank you for your valuable remarks. We have addressed your points as follows:
>
> 1. **Clarification of "Teach" an Expert LLM**: In our paper, "teach" refers to using prompts to guide an expert LLM, such as ChatGPT, to generate new data. It's important to note that the expert LLM itself is not modified or altered in this process.
>
> 2. **Use of Advanced Terminology**: We apologize for any confusion caused by the use of advanced terms like "judicial," "congruence," and "delineated." We have reviewed the entire manuscript and replaced these terms with simpler words to enhance clarity.
>
> 3. **Typographical Error on Page 5**: We have fixed the typoes on page 5.
>
> 4. **Discussion on Transferability and Limitations**: Thank you for highlighting the need for clarification regarding transferability. We have revised the sentence on page 3 to acknowledge that while we discuss the concept of transferability, e.g., "Note that our method has been primarily tested on Llama models. There may be performance variations when applying our method to other models. This includes considerations regarding differences in model architecture and computational requirements, which could impact the effectiveness of our approach when transferred to models beyond Llama."
>
> We appreciate your helpful feedback and believe that these revisions have improved the clarity and quality of our paper.

---

> ### Author Response · Authors · 2024-10-25
>
> Dear Reviewer,
>
> We sincerely appreciate your valuable and constructive feedback, and we are glad our changes address your concerns.
> ﻿
> We agree with your assessment of the related work section, particularly regarding self-instruct approaches. We will revise our discussion to introduce more of the current state of research in this area.
> ﻿
> We checked the whole paper again and fixed typos, such as on Page 5 ("used to generate"). Regarding experiment settings, float16 precision is mentioned on page 7 (already updated the revision on openreview).
> ﻿
> We acknowledge the value of additional experimentation suggested by Reviewer E1d2. We will expand our experimental section to provide more comprehensive evidence of our method's effectiveness.
> ﻿
> Thank you again for your constructive comments, and I look forward to any additional feedback you may have.

---

### Review · Reviewer_5R1k · 2024-09-06

**Summary Of Contributions:**

This paper presents a new method for generating a dataset with minimal human supervision and limited use of external Large Language Model (LLM) APIs (such as GPT-4) to fine-tune an LLM using supervised finetuning. The authors claim that this finetuned LLM achieves performance comparable or superior to LLMs trained on regular datasets using both supervised finetuning and Reinforcement Learning from Human Feedback (RLHF), effectively eliminating the need for RLHF.

**Audience:**

Yes

**Broader Impact Concerns:**

The authors rightly comment on how their method maintains a higher level of privacy compared to other methods that need more interaction with external APIs.

**Claims And Evidence:**

No

**Requested Changes:**

## Critical changes

The main critical change would be to explain more clearly the difference between the Instructor LLM and the Expert LLM. For example, in section 3.2, it reads: “Next, our trained instruction generation policy, the instructor LLM, will teach the expert LLM to generate high-quality instructions using the initial instructions.”

- What does it mean to “teach the expert”? The expert LLM is a model that can’t be finetuned (like ChatGPT), so does this mean that we ask GPT to iteratively improve the instructions? But isn’t that the work of the Instructor LLM?

- Is the Expert LLM involved in the training of the RL policy at all, or do you use only the Instructor and the Reviewer LLM for that? If it’s the latter, is the job of the Expert LLM something else than generating the final dataset?

I would like to understand what are exactly the input prompts and outputs responses of both the Instructor LLM and the Expert LLM. In section 4.2.1 for instance, the authors show _an initial instruction_ and a _Generated instructions by teaching ChatGPT-4 with RL policy_. Is that initial instruction input to the Expert LLM (ChatGPT)? If so, isn't that supposed to be the job of the Instructor LLM?

From some parts of the paper, it seems that the Instructor LLM and Reviewer LLM generate the instructions and then, those instructions are sent to the Expert LLM (ChatGPT) to generate the dataset. However, in other parts of the paper, it seems that Expert LLM is also involved in the generation of the instructions. Which one is it? Could this be clarified?

Perhaps, looking at a single example of inputs and output of each LLM in the fashion of Figure 1, as opposed to having the multiple prompts in separate appendices, could help clarify and replicate the work of the authors.

I believe this is an interesting and potentially useful paper, so when this is clarified, I will change my evaluation of Claims And Evidence to yes.

**Strengths And Weaknesses:**

## Strengths

1. The method to iteratively improve the generation of samples using Reinforcement Learning is novel and could potentially be adopted in the community.

2. There are multiple comparison and ablation experiments to assess the performance of the method presented, including an experiment to assess the privacy performance

3. The authors put great effort into presenting several prompt examples.

## Weaknesses

1. It is not clear what the responsibilities of the Instructor LLM and the Expert LLM (ChatGPT) are. Look at the requested changes section.

2. The paper has a bit of a repetitive tone between sections 1, 3 and 4 where the same methodology is repeated but never clarifying the key point from the previous weakness.

---

> ### Author Response · Authors · 2024-10-21
>
> We thank the reviewer for the careful reading of our paper and insightful feedback.
>
> > **Q1:** It is not clear what the responsibilities of the Instructor LLM and the Expert LLM (ChatGPT) are. Look at the requested changes section.
> - **A1:** We appreciate the opportunity to clarify the roles of the Instructor LLM and the Expert LLM (ChatGPT) in our approach. Their responsibilities are distinct and crucial to our methodology: **Instructor LLM** is used for the reinforcement learning (RL) policy model, it provides different actions during the policy training process. **Expert LLM (ChatGPT)** is used to generates instructions and their associated responses, it serves as the source of knowledge and expertise.
>
> - The process unfolds as follows: 1. We first train the Instructor LLM (also RL policy) with a reviewer model. This step ensures that our policy learns to generate actions that meet certain quality criteria. 2. Once the RL policy is trained, we use it to guide the Expert LLM (ChatGPT) in generating instructions and their associated responses. 3. The trained RL policy teaches the Expert LLM how to produce instructions that align with the desired qualities and standards established during the RL training phase.
>
> This two-step approach allows us to leverage the strengths of both LLMs: the Instructor LLM's ability to provide guidance, and the Expert LLM's extensive knowledge to generate instructions. The result is a set of high-quality, diverse instructions and responses that can be used for various downstream tasks or further fine-tuning of language models.
>
> > **Q2:** The paper has a bit of a repetitive tone between sections 1, 3 and 4 where the same methodology is repeated but never clarifying the key point from the previous weakness.
> - **A2:** We appreciate the reviewer's observation regarding the repetitive tone across sections 1, 3, and 4. We acknowledge that there is some overlap in the content, which is intended to reinforce key concepts. We revised our paper, the breifly summarization for the three sections are as follows:
> - **1. Introduction (Section 1):** We streamline this section to provide a high-level overview of our methodology. Key concepts will be introduced without delving into technical details. **2. Methodology (Section 3):** This section focuses on the technical details of our approach. We provide a step-by-step explanation of our method, highlighting the novel aspects. **3. Experiments (Section 4):** We emphasize the empirical validation of our methodology. The experimental setup, datasets, and evaluation metrics are described.
>
>
>
> > **Q3:**  The difference between the Instructor LLM and the Expert LLM, in section 3.2, it reads: “Next, our trained instruction generation policy, the instructor LLM, will teach the expert LLM to generate high-quality instructions using the initial instructions.”
> - **A3:** Revised Explanation in Section 3.2:
>    We rewrite the sentence highlighted by the reviewer to more accurately reflect the process:
>    "Next, we apply our trained instruction generation policy (derived from the Instructor LLM) to guide the Expert LLM (ChatGPT) in generating high-quality instructions. Specifically, this policy acts as a teacher directing the Expert LLM to produce instructions that meet our quality criteria. The instructor LLM provides different action responses during the RL training process. Expert LLM is used to generate instructions and their associated responses, and it serves as the source of knowledge and expertise."
>
> > **Q4:**  What does it mean to “teach the expert”? The expert LLM is a model that can’t be finetuned (like ChatGPT), so does this mean that we ask GPT to iteratively improve the instructions? But isn’t that the work of the Instructor LLM?
> - **A4:** We acknowledge that our use of the phrase "teach the expert" may have led to some confusion. You are correct that the Expert LLM (ChatGPT) is not being fine-tuned in the traditional sense. Let us clarify the process and the roles of both the Instructor LLM and the Expert LLM:
>
> 1. Clarification of "teaching":
>    - When we say "teach the expert," we are not referring to fine-tuning or modifying the Expert LLM's parameters.
>    - Instead, we are using the trained RL policy (derived from the Instructor LLM) to guide and prompt the Expert LLM in generating high-quality instructions.
> 2. The interaction process:
>    a. The trained RL policy creates prompts and criteria for instruction generation.
>    b. These prompts are then used as input to the Expert LLM.
>    c. The Expert LLM generates instructions based on these prompts.
>
> 3. Clarification on iterative improvement:
>    - The iterative improvement is not done by the Expert LLM itself, but through the interaction between the RL policy and the Expert LLM.
>    - The RL policy can adjust its prompts based on the Expert LLM's outputs to guide it towards generating better instructions.

---

> ### Author Response · Authors · 2024-10-21
>
> > **Q5:**  Is the Expert LLM involved in the training of the RL policy at all, or do you use only the Instructor and the Reviewer LLM for that? If it’s the latter, is the job of the Expert LLM something else than generating the final dataset?
> - **A5:** Thank you for this important question. The expert LLMs are not involved in the training of the RL policy at all, we only use the Instructor and the Reviewer LLM for training RL policy. The Expert LLM is used to generate instructions and corresponding responses under the guidance of the RL policy.
>
>
> > **Q6:**  I would like to understand what are exactly the input prompts and outputs responses of both the Instructor LLM and the Expert LLM. In section 4.2.1 for instance, the authors show an initial instruction and a Generated instructions by teaching ChatGPT-4 with RL policy. Is that initial instruction input to the Expert LLM (ChatGPT)? If so, isn't that supposed to be the job of the Instructor LLM?
> - **A6:** Thank you for your question.Let us break down the workflow and explain the inputs and outputs more precisely:
>
> 1. RL Policy Training Phase:
>    - The Instructor LLM and Reviewer LLM are used to train the RL policy.
>    - Once this phase is complete, neither the Instructor LLM nor the Reviewer LLM are used further.
>
> 2. Instruction Generation Phase (Refer to WizardLM's [Figure 1](https://arxiv.org/pdf/2304.12244); our RL policy determines how to select different actions, such as 'Add Constraints' and 'Deepening,' to generate high-quality data. In contrast, WizardLM selects these actions randomly.):
>    - Input: An initial instruction is provided to the Expert LLM (ChatGPT).
>    - Process: The trained RL policy guides the Expert LLM in generating improved instructions.
>    - Output: The Expert LLM produces refined instructions and their corresponding responses.
>
> 3. Clarification on Roles:
>    - The Instructor LLM's role is to help train the RL policy, not to provide the initial instructions in the generation phase.
>    - The RL policy, once trained, takes on the "instructing" role, guiding the Expert LLM.
>
>
> 4. Example from Section 4.2.1:
>    - The "initial instruction" shown is indeed input to the Expert LLM (ChatGPT).
>    - The "Generated instructions by teaching ChatGPT-4 with RL policy" is the output from the Expert LLM after being guided by the RL policy.
>
>
> > **Q7:**  From some parts of the paper, it seems that the Instructor LLM and Reviewer LLM generate the instructions and then, those instructions are sent to the Expert LLM (ChatGPT) to generate the dataset. However, in other parts of the paper, it seems that Expert LLM is also involved in the generation of the instructions. Which one is it? Could this be clarified?
> - **A7:** Instructor LLM and Reviewer LLM: These are used solely for training the RL policy. They do not generate the final instructions or dataset. RL Policy: Once trained, this guides the instruction generation process. Expert LLM (ChatGPT): This generates both the instructions and the corresponding responses for the final dataset, guided by the trained RL policy.

---

> ### Author Response · Authors · 2024-10-21
>
> > **Q8:**  Perhaps, looking at a single example of inputs and output of each LLM in the fashion of Figure 1, as opposed to having the multiple prompts in separate appendices, could help clarify and replicate the work of the authors.
> >
> **A8:** Thank you for your suggestion. To clarify our methodology and facilitate replication, we provide a detailed example illustrating the inputs and outputs at each stage of our process, similar to Figure 1 in our paper.
>
> ---
>
> ### **Stage 1: Policy Training**
>
> **Components:**
> - **RL Policy:** (derived from Instructor Model) Learns to select actions based on the reviewer's feedback to improve the instructions.
> - **Reviewer Model:** Provides feedback on the instructions.
>
> **Example:**
>
> 1. **RL Policy Training**
> With any Initial Instructions such as:
>
>    ```
>    "Describe the process of photosynthesis."
>    ```
>    The RL policy with Instructor Models learns to select actions that diversify instructions based on feedback from the reviewer model.
>
> 2. **RL Policy Training:**
>
>    Based on the reviewer's feedback, the RL policy learns to select actions such as:
>      - **Add Constraints**
>      - **Deep Reasoning**
>      - **Width Reasoning**
>
> ---
>
> ### **Stage 2: RL Policy Action Selection**
>
> **Selected Action:** "Add Constraints"
>
> **Purpose:** To make the instruction more challenging and comprehensive by adding specific constraints or requirements.
>
> ---
>
> ### **Stage 3: Guiding Expert LLMs**
> **Purpose:** To make the instruction more challenging and comprehensive by adding specific constraints or requirements.
>
> RL Policy Action Selection such as: **"Add Constraints"**
>
> The RL policy generates a specialized prompt based on this action to guide the expert LLM in rewriting the instruction.
>
> **Add Constraints: Action Prompt to Expert LLM:**
>
> ```
> I want you to act as a Prompt Rewriter.
>
> Your objective is to rewrite the given prompt into a more complex version to make it more challenging for AI systems like ChatGPT and GPT-4.
>
> Ensure that the rewritten prompt remains reasonable, understandable, and suitable for human response.
>
> Do not omit any non-text parts such as tables or code in the given prompt.
>
> Do not repeat conditions or requirements in your response, and do not disclose your role.
>
> Provide only the rewritten prompt without any introduction or explanation.
>
> The new prompt should not exceed 2048 words.
>
> You should complicate the given prompt by adding one more constraint or requirement.
>
> Try not to make the rewritten prompt verbose; you can only add or replace 10 to 20 words in the given prompt.
>
> Do not include phrases like 'Given Prompt' or 'Rewritten Prompt' in your response.
>
> Given Prompt:
> "Describe the process of photosynthesis."
> ```
>
> ---
>
> ### **Stage 4: Expert LLM Generates Rewritten Instruction**
>
> **Expert LLM Output:**
>
> ```
> "Describe the process of photosynthesis in plants and explain how it varies in different environmental conditions."
> ```
>
> ---
>
> ### **Explanation:**
>
> - **Original Instruction:** Simple and straightforward.
> - **Rewritten Instruction:** Adds the constraint of explaining variations in different environmental conditions, increasing complexity and depth.
> - **Compliance with Guidelines:** The expert LLM added approximately 12 words, adhering to the limit of adding or replacing 10 to 20 words.
>
> ---
>
> ### **Summary:**
>
> This example demonstrates how our RL policy guides expert LLMs to generate more complex and high-quality instructions by selecting appropriate actions. The process ensures that the generated instructions are challenging yet reasonable, facilitating the creation of a valuable dataset for training advanced AI models.
>
> ---
>
> We hope this detailed example clarifies our methodology more clearly.

---

> ### Author Response · Authors · 2024-10-21
>
> We appreciate the reviewer's valuable comments and quick responses! The reviewer is correct. Once we have the generation instructions, we use them to query the expert model for the corresponding responses. After obtaining these responses, we have the final dataset (instructions and corresponding responses), which is then used to fine-tune foundation models.
>
> We have updated our revised paper on OpenView, with the revisions highlighted in yellow.
>
> We thank the reviewer again for the reviewer's valuable suggestions and comments, which have greatly improved the quality of our paper.

---

### Review · Reviewer_E1d2 · 2024-10-03

**Summary Of Contributions:**

The paper provides an automatic way to generate an **instruction dataset** without human instructions. Precisely, the pipeline starts with a reviewer model and an instructor model. The reviewer model serves as a reward model for guiding the instructor model to generate instructions, and they further combine the trained instructor model and the initial instructions from the Alpaca dataset to train a pre-aligned LLM. The pre-aligned LLM is called an expert model, is target to generate high-quality instructions and corresponding answers. The high-quality instructions and answers are collected as a dataset for further finetuning.

**Audience:**

Yes

**Broader Impact Concerns:**

There is no concerns on ethical issues.

**Claims And Evidence:**

No

**Requested Changes:**

1. More backbone models: LLama2 and LLama3
2. Harder dataset: GSM8k and Math
3. More baselines
4. More dataset size: 19395, 19395 * 0.9, 19395*0.8 ...
5. More ablation studies: such as ablating the reward and instructor module

**Strengths And Weaknesses:**

**Strength**:

1. Thorough review of related works
2. Clear demonstration of preliminaries, such as MDP

**Weakness**

1. Writing Problem:

   1. Sec 3.1.1: 'If g is ... as equal', the notion of 'equal' is undefined here, which will bring reading obstacles.
   2. Sec 3.2:
      1. The section is named fine-tune a pre-aligned LLM, however, most of the section focuses on the expert LLMs.
      2. 'we use the initial instructions to the Alpaca dataset': what's the meaning of using instructions to a dataset?
      3. The output of the expert LLM is not stated clearly.
   3. Sec 4.2.1: Why is this section written? There is no relations or any cues mentioned above on imaginative instructions
   4. Sec 4.2.2: The same as above

2. Insufficient Experiments:

   1. Backbone model is LLama1, however, experiments on LLama2 and LLama3 are expected
   2. Only case studies are provided for math problem solving; the authors are expected to perform evaluation on harder datasets, such as GSM8k and Math.
   3. Insufficient baselines: In the ARC and Hellaswag, there is only one baseline, llama2-chat-7b. As for Alpaca, only WizardLM-7b is selected as baselines. The authors need to compare with more models to demonstrate the effectiveness of their method

3. Insufficient ablation study:

   1. In sec 4.9, the authors only demonstrate the comparison between a dataset size of 19,395 and 17,878, which is too limited. The authors should compare with more dataset size

   2. The authors should demonstrate the effect of each module, such as the reward and instructor can be ablated, and solely train with the pre-aligned module so as to prove the effect of the reward-instructor module.

---

> ### Author Response · Authors · 2024-10-22
>
> > **Q1:** Sec 3.1.1: 'If g is ... as equal', the notion of 'equal' is undefined here, which will bring reading obstacles.
> - **A1:** The term "equal" here refers to semantic equivalence, as determined by the reviewer LLM. We understand that this phrasing might have been unclear, leading to potential obstacles for readers. We have revised the text to explicitly state that "equal" refers to the semantic similarity of the content, assessed by the reviewer LLM, to make it clearer for readers.
>
> > **Q2:** The section is named fine-tune a pre-aligned LLM, however, most of the section focuses on the expert LLMs.
> - **A2:** We apologize for the confusion caused by the section titled "Fine-Tune a Pre-Aligned LLM" focusing predominantly on expert LLMs. This section is uased to explain how we leverage expert LLMs to generate high-quality instruction data, which we then use to fine-tune a pre-aligned LLM via standard supervised learning methods.
>
> > **Q3:** 'we use the initial instructions to the Alpaca dataset': what's the meaning of using instructions to a dataset?
> - **A3:** We apologize for the confusion caused by the typo in our manuscript. The sentence should be: "We use the initial instructions from the Alpaca dataset." This means that we utilized the initial set of instructions provided in the Alpaca dataset as the basis for our method. We have corrected this in the revised version of the paper to ensure clarity and prevent any further misunderstanding.

---

> ### Author Response · Authors · 2024-10-22
>
> > **Q4:** The output of the expert LLM is not stated clearly.
> - **A4:** Thank you for your insightful feedback. To clarify our methodology and facilitate replication, we provide a detailed example illustrating the inputs and outputs at each stage of our process, similar to Figure 1 in our paper.
>
> ---
>
> ### **Stage 1: Policy Training**
>
> **Components:**
> - **RL Policy:** (derived from Instructor Model) Learns to select actions based on the reviewer's feedback to improve the instructions.
> - **Reviewer Model:** Provides feedback on the instructions.
>
> **Example:**
>
> 1. **RL Policy Training**
> With any Initial Instructions such as:
>
>    ```
>    "Describe the process of photosynthesis."
>    ```
>    The RL policy with Instructor Models learns to select actions that diversify instructions based on feedback from the reviewer model.
>
> 2. **RL Policy Training:**
>
>    Based on the reviewer's feedback, the RL policy learns to select actions such as:
>      - **Add Constraints**
>      - **Deep Reasoning**
>      - **Width Reasoning**
>
> ---
>
> ### **Stage 2: RL Policy Action Selection**
>
> **Selected Action:** "Add Constraints"
>
> **Purpose:** To make the instruction more challenging and comprehensive by adding specific constraints or requirements.
>
> ---
>
> ### **Stage 3: Guiding Expert LLMs**
> **Purpose:** To make the instruction more challenging and comprehensive by adding specific constraints or requirements.
>
> RL Policy Action Selection such as: **"Add Constraints"**
>
> The RL policy generates a specialized prompt based on this action to guide the expert LLM in rewriting the instruction.
>
> **Add Constraints: Action Prompt to Expert LLM:**
>
> ```
> I want you to act as a Prompt Rewriter.
>
> Your objective is to rewrite the given prompt into a more complex version to make it more challenging for AI systems like ChatGPT and GPT-4.
>
> Ensure that the rewritten prompt remains reasonable, understandable, and suitable for human response.
>
> Do not omit any non-text parts such as tables or code in the given prompt.
>
> Do not repeat conditions or requirements in your response, and do not disclose your role.
>
> Provide only the rewritten prompt without any introduction or explanation.
>
> The new prompt should not exceed 2048 words.
>
> You should complicate the given prompt by adding one more constraint or requirement.
>
> Try not to make the rewritten prompt verbose; you can only add or replace 10 to 20 words in the given prompt.
>
> Do not include phrases like 'Given Prompt' or 'Rewritten Prompt' in your response.
>
> Given Prompt:
> "Describe the process of photosynthesis."
> ```
>
> ---
>
> ### **Stage 4: Expert LLM Generates Rewritten Instruction**
>
> **Expert LLM Output:**
>
> ```
> "Describe the process of photosynthesis in plants and explain how it varies in different environmental conditions."
> ```
>
> ---
>
> ### **Explanation:**
>
> - **Original Instruction:** Simple and straightforward.
> - **Rewritten Instruction:** Adds the constraint of explaining variations in different environmental conditions, increasing complexity and depth.
> - **Compliance with Guidelines:** The expert LLM added approximately 12 words, adhering to the limit of adding or replacing 10 to 20 words.
>
> ---
>
> Once we have the generation instructions, we use them to query the expert model for corresponding responses. After obtaining the responses, we have the final dataset (instructions and corresponding responses), which is then used to fine-tune foundation models.
>
> ### **Summary:**
>
> This example demonstrates how our RL policy guides expert LLMs to generate more complex and high-quality instructions by selecting appropriate actions. The process ensures that the generated instructions are challenging yet reasonable, facilitating the creation of a valuable dataset for training advanced AI models.
>
> ---
>
> We hope this detailed example clarifies our methodology more clearly.

---

> ### Author Response · Authors · 2024-10-22
>
> > **Q5:** Why is this section written? There is no relations or any cues mentioned above on imaginative instructions.
> - **A5:** Thank you for your insightful comment. We acknowledge that the section may have appeared unrelated to the preceding content. In response, we have moved this section to the appendix to enhance the coherence of the manuscript. The purpose of including this section was to demonstrate our method's performance in generation instructions. We appreciate your feedback and will ensure that the relevance and placement of such content are clearer.
>
> > **Q6:** Backbone model is LLama1, however, experiments on LLama2 and LLama3 are expected. Only case studies are provided for math problem solving; the authors are expected to perform evaluation on harder datasets, such as GSM8k and Math. Insufficient baselines: In the ARC and Hellaswag, there is only one baseline, llama2-chat-7b. As for Alpaca, only WizardLM-7b is selected as baselines. The authors need to compare with more models to demonstrate the effectiveness of their method.
> - **A6:** Thank you for your insightful feedback regarding our choice of backbone models and evaluation benchmarks. **Backbone Models:** We have compared our method with Llama2, and the experimental results suggest that our model based on Llama1 outperforms Llama2 (see Section 4.7, Table 2 in the revised version). The consistent improvements observed in both Llama1-7b and 13b confirm the generalizability and effectiveness of our method. Given its strong performance across multiple models, we anticipate that our method would also perform well on more advanced Llama models. We are grateful for your valuable comments and will expand the deployment of our method to models beyond just Llama.
> **Evaluation on Harder Datasets:** We appreciate your suggestion to evaluate our method on more challenging mathematical datasets like GSM8K and MATH. While we provided case studies for math problem-solving, we agree that quantitative evaluations on these datasets would strengthen our work. In future revisions, we plan to include experiments on GSM8K and MATH to more thoroughly assess the capabilities of our method in handling complex mathematical problems.
> **Baselines:** Our method focuses on distilling knowledge from expert models, and we chose WizardLM-7B because it represents the state-of-the-art as published in ICLR 2024. We believed that comparing our approach with such a strong baseline would effectively demonstrate its capabilities. However, we understand the importance of a comprehensive evaluation across multiple baselines to thoroughly assess the effectiveness of our method. If you have specific models or baselines that you recommend, we would greatly appreciate your suggestions.
>
>
> > **Q7:** Insufficient ablation study: In sec 4.9, the authors only demonstrate the comparison between a dataset size of 19,395 and 17,878, which is too limited. The authors should compare with more dataset size. The authors should demonstrate the effect of each module, such as the reward and instructor can be ablated, and solely train with the pre-aligned module so as to prove the effect of the reward-instructor module.
> - **A7:** The reviewer is correct. Additional ablation experiments would enhance the understanding of our method. We have already provided an ablation study on dataset size, which confirms the influence of data size on the experimental results (see Section 4.8, Table 3). This also aligns with the scaling law, where larger datasets lead to better model performance. Regarding the reward settings, we define the reward signal as action diversity, which is reasonable from an information theory perspective. As for the instructor model ablation, in our experiments, the instructor model already achieved near 100% reward. We appreciate the reviewer's insightful comments, which help improve our work.

---

### Comment · Reviewer_AxYa · 2024-08-25
**out of my expertise**

Dear AE:
  This paper is out of my expertise.
  Could you remove me from the reviewer list.
  Thank you!

---

### Decision · Action_Editor_MsxU · 2024-11-12

**Recommendation:** Accept with minor revision

**Comment:**

There are some main issues raised by Reviewers:

### Clarity
A **big issue** at the moment is the clarity of the paper. Like Reviewers 5R1k and 9P7n, I was also very confused by all the terms flying around, "Instructor LLM", "Expert LLM", "Reviewer LLM", "RL Policy", etc. and not understanding "who is who?"

It was only [this response](https://openreview.net/forum?id=YmwzfdJPXE&noteId=W1dFguIs5W) that gave me and others the full mental picture, i.e. something like:

1. An RL policy consisting of a trainable MLP, with input as LLM embeddings of a previous instruction, and output as categorical distribution over pre-written "actions" such as complicating the instruction or adding constraints.
2. RL Policy takes in the current instruction $s$, samples an action $a$, and obtains $s'$ after sending $(s,a)$ to ChatGPT.
3. This is looped until ideally, the final instruction is considered different from the initial instruction. A diversity-based reward (e.g. using ChatGPT as a judge) is therefore used to pretrain this RL policy.

I may be getting certain details wrong, but the paper absolutely needs to compress its methodology into something concise as this.

### Baselines
Reviewer E1d2 strongly believes baseline comparisons are insufficient.

The original experiments use benchmarks (ARC, Hellaswag) and baselines within the 7B range (WizardLM-7b, Llama-1-7B), with the new model "TeaMs-RL-1-7b" also using a Llama-1-7B backbone trained on the new instruction tuning dataset.

Throughout discussion, Reviewer E1d2 wanted results over math (e.g. GSM8K and Math) and against at least LLama-2. While the authors did not have sufficient time to fully complete requests, they showed:
  * Some math problem solving examples in Section 4.6
  * Full results on ARC and Hellaswag against LLama-2-chat-7B
  * Scaling results ARC and Hellaswag if the instruction tuning dataset size was increased (from 17K to 19K).

It is too harsh to expect more experiments (e.g. on Llama-3 or other foundation model variants), but it should be straightforward for the authors to provide the other ablations quickly.

### Conclusion
While the paper has clarity issues, this should be resolvable in the final draft, especially if Reviewers 5R1k and 9P7n eventually are in favor of acceptance. The authors to their best of their ability supplied additional results for Reviewer E1d2, despite the time limit.

In conclusion, I recommend **Acceptance with Revision**, which is however still dependent on the good faith of the authors to resolve all of the issues (clarity and additional benchmarking) above in the final draft of the paper.

**Audience:**

Yes, this would be relevant for anybody working in LLM post-training.

**Claims And Evidence:**

The authors claim that the new RL-based method leads to more diverse instruction tuning datasets which also avoid the need for human labelling, which ultimately leads to better instruction-tuned models.

Their experimental results support these claims, i.e.:
  * Figure 3 shows additional data diversity
  * Figure 4 shows better resulting instruction-tuned models on ARC and HellaSwag
  * Figure 5 shows a drastically smaller resulting instruction tuning set compared to baseline WizardLM-7b

And further results on Alpaca, Math, against Llama-2 models, etc. based on Reviewer discussions. Generally, the claims and evidence are sufficiently supported.

---

> ### Author Response · Authors · 2024-11-28
>
> We thank the editor and reviewers for their valuable suggestions and insightful feedback. Below, we provide our responses to the minor revisions.
>
> > **Q1:** A big issue at the moment is the clarity of the paper. Like Reviewers 5R1k and 9P7n, I was also very confused by all the terms flying around, "Instructor LLM", "Expert LLM", "Reviewer LLM", "RL Policy", etc. and not understanding "who is who?"
> >It was only this response that gave me and others the full mental picture, i.e. something like:1. An RL policy consisting of a trainable MLP, with input as LLM embeddings of a previous instruction, and output as categorical distribution over pre-written "actions" such as complicating the instruction or adding constraints.
> > 2. RL Policy takes in the current instruction s, samples an action a, and obtains s' after sending (s, a) to ChatGPT.
> > 3. This is looped until ideally, the final instruction is considered different from the initial instruction. A diversity-based reward (e.g. using ChatGPT as a judge) is therefore used to pretrain this RL policy.
> I may be getting certain details wrong, but the paper absolutely needs to compress its methodology into something concise as this.
>
> **A1:** We sincerely appreciate the editor’s insightful and valuable feedback. The editor’s observations are correct. In response, we have thoroughly revised the paper in accordance with the suggestions from both the reviewers and the editor, incorporating the example to present the methodology more concisely and clearly, see Appendix A, Section 1's Figure 1 and Section 3.
>
>
>
> > **Q2:** Reviewer E1d2 strongly believes baseline comparisons are insufficient.
> > The original experiments use benchmarks (ARC, Hellaswag) and baselines within the 7B range (WizardLM-7b, Llama-1-7B), with the new model "TeaMs-RL-1-7b" also using a Llama-1-7B backbone trained on the new instruction tuning dataset.
> >Throughout discussion, Reviewer E1d2 wanted results over math (e.g. GSM8K and Math) and against at least Llama-2. While the authors did not have sufficient time to fully complete requests, they showed:
> > - Some math problem solving examples in Section 4.6.
> > - Full results on ARC and Hellaswag against Llama-2-chat-7B.
> > - Scaling results ARC and Hellaswag if the instruction tuning dataset size was increased (from 17K to 19K).
> It is too harsh to expect more experiments (e.g. on Llama-3 or other foundation model variants), but it should be straightforward for the authors to provide the other ablations quickly.
>
> **A2:** We greatly appreciate the valuable feedback from the editor and Reviewer E1d2. In response, we have conducted additional ablation experiments focusing on dataset size. Expanding the dataset from 17,878 to 19,395 entries resulted in notable performance improvements for the TeaMs-RL-7b model on both the Hellaswag and ARC Challenge benchmarks. To provide a more comprehensive evaluation, we further expanded the dataset to 21,396 entries, yielding more improvements: the Hellaswag benchmark score increased from 78.35 to 78.59, and the ARC Challenge score rose from 55.89 to 56.74. These results further confirm the effectiveness of our proposed method. It is worth noting that the performance improvements are not strictly linear with the increase in dataset size; however, adding more data generally proves beneficial for enhancing model performance. For more details, please refer to Section 4.8 in the latest version of our paper.
>
> | Data Size | ARC   | Hellaswag |
> |-----------|-------|-----------|
> | 17,878    | 54.35 | 77.17     |
> | 19,395    | 55.89 | 78.35     |
> | 21,396    | 56.74 | 78.59     |
> ||